# Learning Fine-grained Parameter Sharing via Sparse Tensor Decomposition

**Cem Üyük**[*]                                                   *cem.ueyuek@tum.de*
*Department of Computer Science*
*Technical University of Munich*
*Bavaria, Germany*

**Mike Lasby**                                                   *mklasby@ucalgary.ca*
*Department of Electrical and Software Engineering*
*Schulich School of Engineering, University of Calgary*
*Calgary, AB, Canada*

**Mohamed Yassin**                                           *mohamed.yassin@ucalgary.ca*
*Department of Electrical and Software Engineering*
*Schulich School of Engineering, University of Calgary*
*Calgary, AB, Canada*

**Utku Evci**[†][*]                                               *evcu@google.com*
*Google DeepMind, Canada*

**Yani Ioannou**[†]                                               *yani.ioannou@ucalgary.ca*
*Department of Electrical and Software Engineering*
*Schulich School of Engineering, University of Calgary*
*Calgary, AB, Canada*

**Reviewed on OpenReview:** *https://openreview.net/forum?id=vbS7Z8Zswe*

## Abstract

Large neural networks achieve state-of-the-art performance on many tasks, yet their sheer size hinders deployment on resource-constrained devices. Among existing compression approaches, cross-layer parameter sharing remains relatively unexplored for transformer models. In this paper, we introduce Fine-grained Parameter Sharing (FiPS), a unified framework for compressing transformer Multi-Layer Perceptrons (MLPs) that combines cross-block parameter sharing, low-rank factorization, and sparsity in a single optimization. FiPS concatenates MLP weight matrices across a group of transformer blocks and factorizes them into a shared basis and sparse, layer-specific projection matrices. Both factors are initialized via singular value decomposition (SVD) and jointly optimized by block-wise reconstruction error minimization. FiPS compresses Vision Transformers (ViTs) by up to 33% with less than 1% top-1 accuracy loss on ImageNet-1k, and by up to 57% when combined with fine-tuning. It also compresses Large Language Models (LLMs) by up to 20% while outperforming existing SVD-based methods in perplexity and downstream benchmarks at matched compression. Combined with Quantization-Aware Training (QAT), 3-bit FiPS on GEMMA-2-2B achieves lower perplexity than 2-bit QAT alone while matching the same $8\times$ compression. These results establish fine-grained parameter sharing as a practical and effective approach for transformer MLP compression.

---

[*]Correspondence to: Cem Üyük <cem.ueyuek@tum.de>, Utku Evci <evcu@google.com>.
[†]Equal advising.

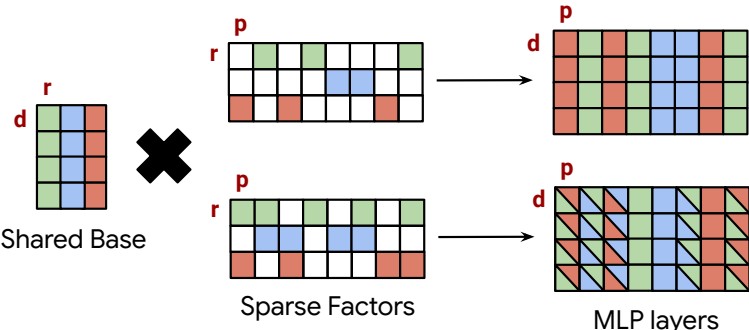

Figure 1: **Fi**ne-grained **P**arameter **S**haring (FiPS).

# 1 Introduction

Over the past decade, large neural networks have delivered impressive performance through scaling of datasets and model sizes. However, this trend has introduced substantial computational, memory, and storage burdens, highlighting the need for efficient model compression to reduce overhead and enable deployment on resource-constrained devices such as mobile phones and embedded systems. In response, researchers have explored various strategies, including tensor decomposition, quantization, distillation, sparsity, adaptive computing methods, and *parameter sharing* (Cheng et al., 2020). While most of these techniques are well-studied and widely adopted, parameter sharing in neural networks remains underexplored and has not yet been leveraged successfully to compress Transformer models, despite its potential for significant parameter-count reduction. Transformer architectures are a particularly natural target for parameter sharing because they are composed of repeated homogeneous blocks whose Multi-Layer Perceptron (MLP) modules have identically shaped fully connected (FC) layers across depth, enabling direct concatenation and joint factorization of weights across blocks.

Sharing parameters across layers reduces memory usage and improves cache efficiency. Prior works have investigated reusing entire Transformer blocks (Lan et al., 2020; Takase & Kiyono, 2023; Lin et al., 2023), yielding more efficient models. Although directly sharing unmodified weights is promising, we hypothesize that a more fine-grained approach could yield superior compression.

We therefore focus on sharing neurons across layers by introducing a shared basis, with each neuron expressed as a linear combination of basis vectors via a projection matrix. We, then, find that enforcing sparsity in the projection matrix is essential for the effectiveness of this approach. This insight leads to our novel parameter sharing algorithm, **Fi**ne-grained **P**arameter **S**haring (FiPS), which, as we demonstrate, effectively compresses large Vision Transformers (ViTs) and Large Language Models (LLMs)[1]. Our contributions include:

- **Systematic Analysis of Cross-Block MLP Sharing.** We systematically explore strategies for sharing bases and neurons across transformer MLP modules, examining sharing granularity, concatenation schemes, and sparsity patterns. We identify when neuron sharing is most effective and how global versus local and structured versus unstructured sparsity interact with cross-block sharing.

- **FiPS Algorithm[2].** A shared basis and sparse layer-specific projection matrices are initialized via SVD and jointly optimized by block-wise reconstruction error minimization, followed by optional end-to-end fine-tuning (FT).

- **State-of-the-Art ViT & LLM Compression.** FiPS delivers substantial compression with minimal performance loss, surpassing recent baselines. It compresses DEIT-B and SWIN-L by up to 33% with <1% top-1 accuracy drop across five vision benchmarks (up to 57% with fine-tuning), and compresses LLAMA-7B and LLAMA-3.1-8B by up to 20% while outperforming existing SVD-based methods on 10 NLP benchmarks.

---

[1]All model links are available in § A.3.
[2]Code: https://github.com/cemuyuk/FiPS.

- **Quantization-Aware Training (QAT).** We demonstrate that 3-bit QAT combined with FiPS compresses GEMMA-2-2B effectively, achieving the same $8\times$ compression as 2-bit quantization but with markedly better language modeling. This confirms that FiPS is orthogonal to QAT.

The remainder of this paper is organized as follows: Section 2 formalizes the integration of low-rank factorization and sparsity for parameter sharing; Section 3 details the FiPS algorithm; Section 4 presents our empirical findings; and Section 5 provides ablation studies on key design choices. Extensive supplementary material is provided in the Appendix, including additional experiments, algorithmic details, and hyperparameter sweeps.

## 2 Parameter Sharing Through Sparse Tensor Decomposition

Consider a weight matrix $\mathbf{W} \in \mathbb{R}^{d \times p}$, where $p > d$, that projects feature vectors from a $d$-dimensional space to a $p$-dimensional space with neurons represented by the columns of $\mathbf{W}$. Our objective is to share weights among a subset of these $p$ neurons, reducing the number of unique neurons to $r < p$. In other words, only $r$ columns of $\mathbf{W}$ will contain unique values. These $r$ unique neurons are represented using a shared basis $\mathbf{U} \in \mathbb{R}^{d \times r}$, which is orthogonal at initialization (via SVD) when $r \le d$. The original matrix $\mathbf{W}$ is then reconstructed by mapping each of its $p$ columns to an $r$-dimensional one-hot vector via a projection matrix $\mathbf{V} \in \mathbb{R}^{r \times p}$, i.e., $\mathbf{W} = \mathbf{UV}$. This "one-hot" approach is illustrated in the upper part of Figure 1. However, limiting the number of unique neurons to $r$ constrains the representational capacity of $\mathbf{W}$. To address this limitation, we can increase the number of non-zero elements in $\mathbf{V}$, effectively creating linear combinations of basis neurons and generating a significantly larger set of unique neuron representations, as shown in the lower part of Figure 1.

This approach readily extends from sharing neurons within a single weight matrix $\mathbf{W}$ to multiple weight matrices $\mathcal{W} = \{\mathbf{W}_1, \ldots, \mathbf{W}_N\}$ via concatenation (see Figure 6). The collection $\mathcal{W}$ is naturally a 3rd-order tensor in $\mathbb{R}^{d \times p \times N}$; our long-axis concatenation corresponds to its mode-2 unfolding, and the resulting factorization into a shared $\mathbf{U}$ and layer-specific sparse $\mathbf{V}_i$ is the order-2 (matrix) case of a Tucker-1 decomposition with sparse core slices—distinct from higher-order methods such as tensor train or CP decomposition. Specifically, fine-grained parameter sharing across multiple layers can be achieved by expanding the projection matrix $\mathbf{V}$ and the shared basis $\mathbf{U}$. Sharing neurons between layers in this manner may be viewed as a factorization of $\mathcal{W}$, where the first factor $\mathbf{U}$ is shared across $N$ layers and the second, layer-specific factor $\mathbf{V}$ is sparse. Existing low-rank decomposition techniques can therefore be employed to obtain a shared basis—orthogonal at initialization via SVD, though not constrained to remain so during optimization—while sparsity in the projection matrices is induced using standard pruning and sparse training methods. When the parameter budget yields $r \le d$, this is a genuine low-rank factorization of $\mathcal{W}$ with an orthogonal basis $\mathbf{U}$ at initialization; at higher budgets, $r$ may exceed $d$, in which case $\mathbf{U}$ acts as an overcomplete shared dictionary that is no longer orthogonal, and parameter reduction arises from sharing and sparsity rather than rank reduction (discussed further in §3).

In the following sections, we investigate optimal layer-tying strategies within our framework, using a 12-block DEIT-B encoder with a single MLP module per block, pretrained on ImageNet-1k (Deng et al., 2009). Specifically, we focus on MLP modules, which account for the majority of parameters (e.g., 70.5% in GEMMA-2-9B (Team et al., 2024)) and comprise two fully connected (FC) layers (i.e., FC-1 and FC-2 for DEIT-B) of dimensions $\mathbb{R}^{d \times p}$ and $\mathbb{R}^{p \times d}$ with $p = 4d$. The "parameter budget" denotes the fraction of nonzero parameters retained after truncated SVD and sparsification; e.g., 25% retains one-quarter of each MLP's weights. We measure the overall compression ratio as the percentage reduction in model size in bits, including sparsity metadata; e.g., 20% compression reduces storage by 20%. Concretely, the compressed model size (in bits) is

$$|\theta_{\text{non-MLP}}| \cdot b \;+\; G \cdot |\mathbf{U}| \cdot b \;+\; \sum_i \big[(1-s) \cdot |\mathbf{V}_i| \cdot b \;+\; |\mathbf{V}_i|\big], \tag{1}$$

where $G$ is the number of groups, $s$ the sparsity level, $b$ the bits per parameter, and the final $|\mathbf{V}_i|$ term accounts for the 1-bit mask per element in unstructured sparsity. The compression ratio is then 1 minus this quantity divided by the original model size $|\theta_{\text{original}}| \cdot b_{\text{original}}$. Note that non-MLP parameters (attention, embeddings, layer norms) remain uncompressed and are counted at full precision (full derivation in Appendix A.2). The rank $r$ of the shared basis is not a free hyperparameter—given a parameter budget $b_\%$, the number of FC

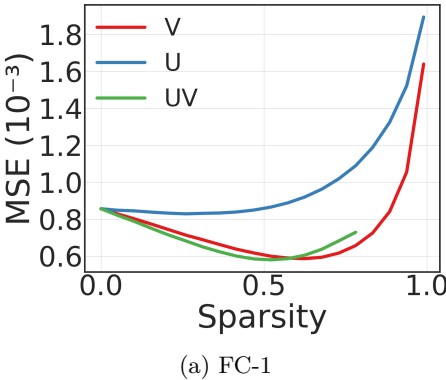 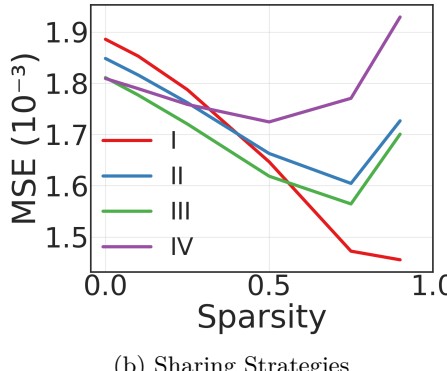

(a) FC-1                                               (b) Sharing Strategies

Figure 2: **Initial Experiments.** (a) Reconstruction error with varying levels of sparsity on different factors of the low-rank decomposition of FC-1 under 25% parameter budget. Results are analogous for FC-2, i.e., inducing sparsity on the larger factor yields a higher rank and, thus, lower reconstruction error. (b) Mean reconstruction error across four FCs of two distinct encoder blocks' MLPs under various parameter sharing schemes and sparsities. See § 2.2 for details.

layers in the group $N$ (number of blocks × FCs per block), and target sparsity $s$, it is uniquely determined by:

$$r = \left\lfloor \frac{b_\% \, N \, d \, p}{d + N(1-s)\, p} \right\rfloor. \tag{2}$$

Conceptually, FiPS differs from standard low-rank compression in three key aspects: (i) the basis is shared across layers rather than learned per layer; (ii) sparsity is imposed on the projection factors rather than the basis, enabling flexible neuron recombination; and (iii) optimization is performed jointly across layers using block-wise reconstruction loss. Together, these components transform low-rank approximation into a fine-grained parameter sharing mechanism.

## 2.1 Optimal Sparsity Allocation Across Factors

Before implementing parameter sharing via shared bases, we decompose individual FC layers of MLP modules using truncated SVD at a 25% parameter budget. Subsequently, sparsity is introduced by pruning low-magnitude values. Specifically, we examine sparsity induction in: (1) $\mathbf{U}$, (2) $\mathbf{V}$, and (3) both $\mathbf{U}$ and $\mathbf{V}$. Throughout this process, we vary the sparsity levels of the matrices while maintaining a constant total number of non-zero parameters. The resulting reconstruction errors are presented in Figure 2a. Our experiments show that the lowest reconstruction errors occur at sparsity levels between 60% and 80%, as confirmed by a sparsity sweep on ImageNet-1k with DEIT-B (Figure 4), particularly when sparsity is imposed on the larger factor matrix $\mathbf{V}$. We attribute this to the higher redundancy in larger matrices, which facilitates more efficient pruning.

## 2.2 Weight Concatenation and Sharing Dimensions

We investigate parameter sharing across multiple layers by analyzing four FC layers drawn from two distinct MLP modules. To align their dimensions, we transpose the second FC layer of each module, representing every layer as $\mathbf{W} \in \mathbb{R}^{d \times 4d}$. We then examine four concatenation strategies for constructing a shared weight block $\mathbf{W}_s$:

(I) **Long-axis concatenation.** All four weight matrices $\mathbf{W}$ are concatenated along their output (neuron) dimension, resulting in $\mathbf{W}_s \in \mathbb{R}^{d \times 16d}$.

(II) **Module-wise long, inter-module short.** Inside each MLP, its two FC layers are first concatenated along the output dimension, producing $\mathbf{W}_s \in \mathbb{R}^{2d \times 8d}$.

(III) **Module-wise short, inter-module long.** Inside each MLP, its two FC layers are first concatenated along the input dimension, again yielding $\mathbf{W}_s \in \mathbb{R}^{2d \times 8d}$.

(IV) **Short-axis concatenation.** All four $\mathbf{W}$ matrices are concatenated along their input (feature) dimension, aligning features rather than neurons and resulting in $\mathbf{W}_s \in \mathbb{R}^{4d \times 4d}$.

For each concatenated block $\mathbf{W}_s$, we perform truncated SVD to retain the top $r$ singular vectors, followed by sparsification of the right singular matrix $\mathbf{V}$ via magnitude pruning (see § 2.1). Reconstruction is then obtained using the resulting shared basis, and mean squared error (MSE) is reported in Figure 2b. Empirically, concatenation along the longer dimension consistently achieves the lowest reconstruction error—particularly under high sparsity—motivating our choice of full long-axis concatenation throughout. Further implementation details are provided in § A.1.

## 2.3 Parameter Sharing Across Layers

This section examines redundancy and interdependencies among MLP modules to identify optimal parameter sharing groupings. We first decompose each module individually at rank $r = 180$ and plot the resulting mean squared error in the lower panel of Figure 3a. The error increases nearly monotonically with module depth, indicating that deeper layers require greater representational capacity. Next, we evaluate pairwise parameter sharing between modules $i$ and $j$ through a shared basis $\mathbf{U}$. Parameter sharing reduces the total number of unique parameters but increases the reconstruction error for each module. We denote the average error increase due to parameter sharing between modules $i$ and $j$ as $MSE_{i,j}^{\downarrow}$:

$$MSE_{i,j}^{\downarrow} = \frac{(MSE_{i,j} - MSE_i) + (MSE_{j,i} - MSE_j)}{2},$$

where $MSE_{i,j}$ denotes the error of module $i$ when sharing a basis with module $j$. Figure 3a shows that adjacent modules exhibit the smallest error increase due to parameter sharing, motivating the practice of grouping consecutive layers.

We then explore the effect of grouping multiple MLP modules. Increasing group size allows a higher rank for the shared basis $\mathbf{U}$, as shown in Figure 3b. This benefit is most pronounced when the projection matrices $\mathbf{V}$ are sparsified. However, a higher rank does not always improve task performance, since $\mathbf{U}$ must capture a larger set of neurons. Figure 3c demonstrates that sharing across four consecutive MLP modules yields the highest post-compression accuracy.

Overall, our results reveal that (i) deeper modules require greater capacity when compressed in isolation—an effect we confirm with global pruning experiments in § 5; (ii) parameter sharing between adjacent layers curbs the rise in reconstruction error; and (iii) there exists an optimal group size that balances basis rank against sparsity. We encode this choice in the *grouping hyperparameter* $\beta = [\beta_1, \ldots, \beta_G]$, an ordered list where $\beta_g$ gives the number of consecutive blocks whose MLP weights are tied in group $g$, with $\sum_{g=1}^{G} \beta_g = L$ (total blocks). For DEIT-B, $\beta = [4, 4, 4]$ achieves the best trade-off; for LLMs a tapered grouping with smaller groups at the network extremes works best (e.g., $\beta = [4, 6, 6, 6, 6, 4]$ for LLAMA-7B). A full sweep is reported in § A.8.1.

## 3 Fine-grained Parameter Sharing

The insights from § 2 motivate FiPS, an efficient cross-block parameter sharing algorithm grounded in sparse tensor decomposition. FiPS comprises three stages:

1. **Shared Initialization.** Tie the FC layers within each MLP group and apply truncated SVD to their concatenation (see Figure 6), yielding a shared basis $\mathbf{U}$ and projection matrices $\{\mathbf{V}_i\}$.
2. **Local Error Minimization.** Using a small calibration dataset $D$ (§ 4.1), optimize $\mathbf{U}$ and each $\mathbf{V}_i$ to minimize the $\ell_2$ discrepancy between original and compressed activations while enforcing target sparsity in $\mathbf{V}_i$.
3. **Global Error Minimization (Optional).** Fine-tune the compressed model end-to-end under a dynamic sparse training regime to recover performance at higher compression ratios.

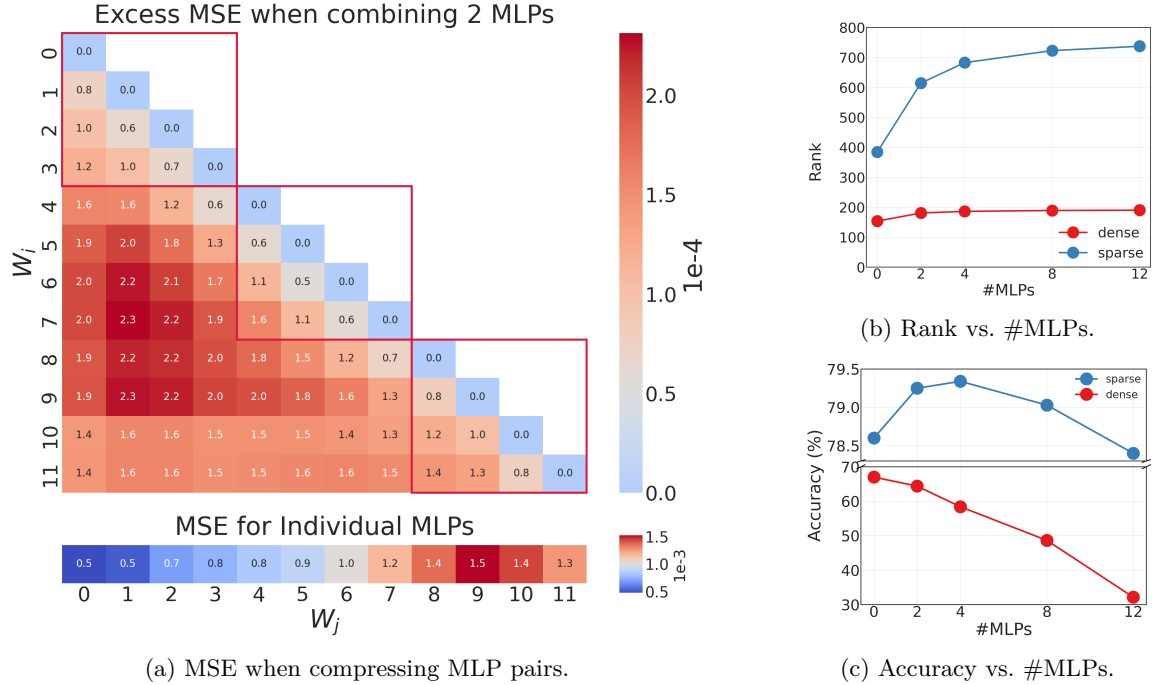

(a) MSE when compressing MLP pairs.

(b) Rank vs. #MLPs.

(c) Accuracy vs. #MLPs.

Figure 3: **Parameter Sharing Groups.** (a top) MSE increases when sharing **U** across different MLP modules; red squares over the diagonal indicate that sharing adjacent modules enhances reconstruction. (a bottom) MSE for compressing individual MLP modules, showing that sharing **U** among consecutive layers typically results in the lowest error. (b) For a fixed parameter budget, the rank of the shared basis **U** stabilizes around four MLP modules, aligning with the optimal group size (c) for maximizing accuracy in DEIT-B, while the dense counterpart continuously worsens.

**Shared Initialization.** We begin by compressing the pretrained model through parameter sharing, achieved by concatenating and decomposing multiple FC layers simultaneously. For higher parameter budgets and sparsity levels (e.g., 50% and 75%, respectively, for DEIT-B), the rank of the shared factor **U** can exceed the model dimension $d$. In this regime, **U** is no longer a low-rank basis but an overcomplete shared dictionary: parameter reduction comes from sharing **U** across blocks and sparsity in **V**, not from rank reduction. The rank $r$ is determined by Equation 2 and remains fixed throughout optimization. To initialize the $k = r - d$ additional dimensions, we compare three strategies: (1) random growth, initializing new neurons in **U** to zero and in **V** via He et al. (2015); (2) neuron splitting, duplicating and halving top neurons following Net2Net (Chen et al., 2016); and (3) hybrid initialization, setting new neurons in **U** to zero and deriving those in **V** from the top-$k$ singular vectors scaled by $1/\tau$, where $\tau > 1$ dampens their initial contribution so they are learned gradually (Evci et al., 2022). After sweeping $\tau$ (see § A.1.1), hybrid initialization outperforms the alternatives by 1–2 percentage points in top-1 accuracy.

Formally, the parameters of a group of FC layers across MLP modules, $\mathbf{W}_1, \mathbf{W}_2, \ldots, \mathbf{W}_N$, are concatenated into $\mathcal{W} = [\mathbf{W}_1; \mathbf{W}_2; \ldots; \mathbf{W}_N]$, where $\mathbf{W}_i \in \mathbb{R}^{d \times p}$.[3] We then apply truncated SVD, $\mathcal{W} = \mathbf{U}\mathbf{\Sigma}\hat{\mathbf{V}}$, to obtain a low-rank approximation, where $\mathbf{U} \in \mathbb{R}^{d \times r}$, $\mathbf{\Sigma} \in \mathbb{R}^{r \times r}$, and $\hat{\mathbf{V}} \in \mathbb{R}^{r \times (N \cdot p)}$. The factor **U** is shared among all layers within the group and remains dense due to its small size. We then multiply $\hat{\mathbf{V}}$ by the singular values to obtain the projection matrix $\mathbf{V} = \mathbf{\Sigma}\hat{\mathbf{V}}$. Finally, the weights are reconstructed as $\mathbf{W}'_i = \mathbf{U}\mathbf{V}_i$, where each $\mathbf{V}_i$ is the slice of **V** corresponding to weight matrix $\mathbf{W}_i$.

**Local Error Minimization.** In the second phase of FiPS, we compute the input and output activations of the original FC layers using a calibration dataset $D$ (described in § 4.1). We use these activations to optimize

---

[3]The second FC layer is transposed to match the dimensions of the first.

---

**Algorithm 1 Fine-grained Parameter Sharing**

---

**Require:** MLP parameters $\mathbf{W}_1, \cdots, \mathbf{W}_N \in \mathbb{R}^{d \times p}$, MLP inputs $\mathbf{A}_i$ and MLP function $\mathbf{f}(\mathbf{W}_i, \mathbf{A}_i)$, Target rank $r$, Learning Rate $\eta$, Steps $\mathbf{T}$.

1: $\mathbf{U}, [\mathbf{V}_1, \mathbf{V}_2, \cdots, \mathbf{V}_N] \leftarrow \textit{TruncatedSVD}([\mathbf{W}_1; \mathbf{W}_2; \cdots; \mathbf{W}_N], k = r)$
2: **for** each training iteration $t = 1$ to $T$ **do**
3: $\quad \mathbf{G_U} = 0$ $\qquad\qquad\qquad\qquad\qquad\qquad\qquad\qquad$ ▷ Gradient accumulator for $\mathbf{U}$
4: $\quad$ **for** each block $i$ **do**
5: $\quad\quad \mathbf{V}_i \leftarrow \textit{Sparsify}(\mathbf{V}_i, t)$ $\qquad\qquad\qquad$ ▷ Potentially increase or adjust sparsity
6: $\quad\quad L_i \leftarrow \textit{MSE\_loss}(\mathbf{f}(\mathbf{W}_i, \mathbf{A}_i), \quad \mathbf{f}(\mathbf{U}\,\mathbf{V}_i, \mathbf{A}_i))$
7: $\quad\quad \mathbf{V}_i \leftarrow \mathbf{V}_i - \eta \nabla_{\mathbf{V}_i} L_i$
8: $\quad\quad \mathbf{G_U} \leftarrow \mathbf{G_U} + \nabla_{\mathbf{U}} L_i$
9: $\quad$ **end for**
10: $\quad \mathbf{U} \leftarrow \mathbf{U} - \dfrac{\eta}{N} \mathbf{G_U}$
11: **end for**
12: **return** $\mathbf{U}, [\mathbf{V}_1, \ldots, \mathbf{V}_N]$

---

the compressed layers by minimizing the $\ell_2$-*loss* between the original and compressed activations:

$$\underset{\mathbf{U}, \mathbf{V_i}, \ldots, \mathbf{V_N}}{\arg\min} \sum_i^N \|\mathbf{W}_i \mathbf{X}_i - \mathbf{U}\mathbf{V}_i \mathbf{X}_i\|_2^2, \tag{3}$$

where $\mathbf{X}_i$ is the inputs to the $i^{\text{th}}$ original FC layer. Sparsity on each $\mathbf{V}_i$ is enforced by the *Sparsify* step of Algorithm 1 rather than as an explicit constraint in the loss; $\mathbf{U}$ is left unconstrained, as orthogonality is only a byproduct of the SVD initialization and need not be preserved.

We explore several sparse training and pruning strategies to identify a sparse $\mathbf{V}$ during optimization: (a) *Static Sparsity*, which fixes the sparsity pattern by retaining the top-magnitude connections before training (Hoefler et al., 2021); (b) *Gradual Magnitude Pruning (GMP)* (Zhu & Gupta, 2017), which progressively increases sparsity by updating the mask every $T$ steps according to the cubic schedule of Kurtic et al. (2023); and (c) *RigL* (Evci et al., 2021), which initializes as in (a) but updates sparse connectivity every $\Delta T$ steps using both gradient and magnitude information. We adopt *GMP* as our final strategy due to its superior performance, achieving up to 4% higher top-1 accuracy on ImageNet-1k compared to the closest baseline. A detailed comparison of sparsification methods is given in Table 5.

During this stage, parameters are shared across multiple MLP modules, allowing gradients to be computed one module at a time. As a result, optimization is significantly more resource-efficient than end-to-end fine-tuning.

**Global Error Minimization.** In this optional stage, we fine-tune the learned parameter sharing scheme end-to-end to further improve performance and leverage the masks learned via GMP. Because the factors $\mathbf{V}_i$ are sparse, we employ the dynamic sparse training method *RigL* during this stage, as it performs slightly better than *Static Sparsity* (see § 4.1).

## 4 Main Results

**Evaluation Metrics.** For ViTs we report top-1 classification accuracy on ImageNet-1k validation. For LLMs we report perplexity (PPL), defined as $\text{PPL} = \exp\left(-\frac{1}{T}\sum_{t=1}^{T} \log p(x_t \mid x_{<t})\right)$, on WikiText-2 and C4, as well as downstream task accuracy via LM-Evaluation-Harness (Sutawika et al., 2023). During compression, reconstruction quality is measured by mean squared error (MSE) between original and compressed activations, i.e., the per-layer $\ell_2$ loss in Equation (3). Lower PPL and MSE indicate better quality; higher accuracy is better.

Table 1: **ViT Compression Results.** ImageNet-1k top-1 validation accuracy of DEIT-B (81.85%) (Touvron et al., 2021) and SWIN-L (86.24%) (Liu et al., 2021) compressed using FiPS and AAFM/GFM across MLP parameter budgets, comparing layer-wise (FiPS) and global error minimization (FiPS+FT). Parameter budgets refer to the fraction of MLP parameters retained (see §2); the corresponding whole-model compression ratios (via Equation 1) are shown in the second row. AAFM/GFM$^\dagger$ results are from Yu & Wu (2023); – denotes unreported metrics.

| Parameter Budget (Comp. Ratio) | 10% (∼57%) | | 25% (∼45%) | | 40% (∼33%) | | 50% (∼25%) | | 75% (∼6%) | |
|---|---|---|---|---|---|---|---|---|---|---|
| Method / Model | DEIT | SWIN | DEIT | SWIN | DEIT | SWIN | DEIT | SWIN | DEIT | SWIN |
| AAFM $^\dagger$ | – | – | – | – | 80.33 | – | 81.21 | 85.04 | 81.76 | 85.94 |
| GFM $^\dagger$ | – | – | – | – | 81.28 | – | 81.62 | 85.44 | 81.83 | 86.01 |
| FiPS (ours) | 70.04 | 74.04 | 80.64 | 84.78 | **81.69** | **85.69** | 81.83 | **85.99** | 81.82 | 86.21 |
| FiPS + FT (ours) | **77.26** | **82.13** | **81.31** | **85.16** | 81.54 | 85.68 | 81.54 | **85.99** | 81.82 | **86.22** |

## 4.1 Vision Transformers

**Experimental Setup.** We evaluate FiPS on DEIT-B (Touvron et al., 2021) and SWIN-L (Liu et al., 2021). Each model is calibrated on 2,560 ImageNet-1k images for 20 epochs, sufficient for convergence; calibration completes in under one hour on a single NVIDIA A6000—comparable to or cheaper than existing post-training compression methods, none of which report full training cost breakdowns. For parameter sharing, we group every four MLP modules in DEIT-B. In SWIN-L, which consists of four stages with 2, 2, 18, and 2 encoder blocks, we share across entire stages for the three smaller ones and use groups of six blocks in the larger stage. Additional hyperparameters are provided in § A.8; sensitivity analyses are in Figure 4.

**ImageNet-1k.** We compare FiPS to the two closest ViT compression baselines that share its post-training, factorization-based setting without distillation or LoRA: Adaptive Atomic Feature Mimicking (AAFM), which compresses output activations rather than weights, and Global Feature Mimicking (GFM), which fine-tunes the compressed network (Yu & Wu, 2023). Both FiPS and FiPS+FT match AAFM and GFM in compute and memory budgets. At a 40% parameter budget, FiPS outperforms AAFM by 1.36 points and GFM by 0.41 points despite GFM's higher cost (Table 1). This pattern holds across all budgets: FiPS consistently achieves the highest accuracy while matching or improving upon the compute and memory budgets of AAFM and GFM. AAFM requires no fine-tuning; GFM requires full end-to-end fine-tuning; FiPS's local error minimization is strictly cheaper than GFM and comparable to AAFM. Notably, a 10% parameter budget corresponds to roughly a 50% compression ratio, where FiPS+FT yields particularly strong gains.

**Transfer Learning.** For transfer learning, we fine-tune for 100 epochs on CIFAR-100, Flowers102, Oxford-III-Pets, and iNaturalist 2019 (Krizhevsky, 2009; Nilsback & Zisserman, 2008; Parkhi et al., 2012; Van Horn et al., 2018), following Yu & Wu (2023). We use *AdamW* (Loshchilov & Hutter, 2019) with learning rates selected from 12 log-spaced values. Models compressed with FiPS transfer significantly better, as shown in Table 2a.

**Latency and Memory Profiling.** Structured sparsity patterns enable efficient hardware implementations with minimal quality impact, as demonstrated in Table 2b using *NMGMP*. With 2:4 structured FiPS, the accuracy degradation remains just above 1% at 10% and 25% MLP parameter budgets; for all other settings, the impact is negligible. We further evaluate FiPS with alternative structured sparsity pruners—*STE*, *SR-STE* (Zhou et al., 2021), and *NMSRigL* (Lee et al., 2023; Lasby et al., 2024)—in Table 7. Latency and memory profiling with the optimal *NMGMP*+FiPS setup leverages NVIDIA's tensor core support for 2:4 sparsity (Mishra et al., 2021) on GPUs and Neural Magic's DeepSparse Engine (Neural Magic, 2021) on CPUs, as shown in Figure 7 and detailed in § A.9.

Table 2: (a) Top-1 accuracy of DEIT-B and compressed variants using GFM and FiPS under different parameter budgets. GFM$^\dagger$ and Original$^\dagger$ results are from Yu & Wu (2023) and Touvron et al. (2021). (b) ImageNet-1k top-1 accuracy of DEIT-B with FiPS at 2:4 structured sparsity (N:M GMP (Lee et al., 2023)) versus 50% unstructured sparsity.

(a)

| | Original$^\dagger$ | GFM$^\dagger$ | | FiPS+RigL FT (ours) | | |
|---|---|---|---|---|---|---|
| P. Budget | 100 | 40% | 50% | 25% | 40% | 50% |
| (Comp. Ratio) | – | (∼33%) | (∼25%) | (∼45%) | (∼33%) | (∼25%) |
| CIFAR-100 | 90.99 | 90.17 | 90.67 | 90.88 | 91.24 | **91.33** |
| Pets | **94.74** | 93.95 | 94.22 | 94.19 | 94.52 | 94.41 |
| Flowers102 | 97.77 | 97.02 | 97.45 | 97.84 | 98.14 | **98.37** |
| iNaturalist 2019 | 77.39 | 77.13 | 77.56 | 77.26 | 77.58 | **77.69** |

(b)

| P. Budget | 10% | 25% | 40% | 50% | 75% |
|---|---|---|---|---|---|
| (Comp. Ratio) | (∼57%) | (∼45%) | (∼33%) | (∼25%) | (∼6%) |
| 2:4 FiPS | 52.36 | 76.88 | 80.59 | 81.31 | 81.51 |
| FiPS | 54.00 | 77.56 | 80.94 | 81.63 | 81.77 |

These results demonstrate that FiPS with structured sparsity translates compression into measurable inference gains on ViTs. Specifically, with a batch size of 64, FiPS with 2:4 sparsity at a 22.14% parameter budget yields a 1.31× speedup on the NVIDIA A4000 and reduces peak VRAM allocation to approximately 0.79× of the original requirement during DEIT-B inference. These gains generalize architecturally: FiPS replaces every dense FC layer $\mathbf{W} \in \mathbb{R}^{d \times p}$ with the product $\mathbf{UV}$, changing the forward pass from a single dense matrix–vector product ($2dp$ FLOPs per token) to a sparse product followed by a dense one ($2rp(1-s) + 2dr$ FLOPs per token, where $s$ is the sparsity level). This FLOP reduction depends only on the factorization parameters $(r, s, d, p)$, not on whether the FC layer resides in a ViT encoder or an LLM decoder. Consequently, the latency and memory improvements measured on DEIT-B are expected to transfer to LLM architectures at matching compression configurations, subject to hardware-specific kernel availability for the relevant dimensions.

## 4.2 Large Language Models

**Experimental Setup.** We evaluate FiPS on three publicly available pretrained LLMs: LLAMA-7B (Touvron et al., 2023), LLAMA-3.1-8B (Grattafiori et al., 2024), and the instruction-tuned GEMMA-2-2B (Team et al., 2024). GEMMA-2-2B is included solely to demonstrate that FiPS is orthogonal to QAT, while the other models are compared against the baselines described below. Unlike ViTs, these LLMs feature three FC layers per MLP in their decoder blocks. For parameter sharing, all FC layers from MLPs within the same group are concatenated. The number of MLPs per group is treated as a hyperparameter and detailed in § A.8.1. Calibration and optimization follow the ViT protocol (described in § 4.1): activations are collected from 8,192 × 20 SlimPajamas (Soboleva et al., 2023) tokens, and block-wise error minimization is run for 40 epochs—without global error minimization (i.e., no full FT)—on a single NVIDIA A100 (80GB), completing within 10 hours per model on a single GPU. To our knowledge, none of the compared baselines report end-to-end compression time; FiPS is the only method in this comparison that provides explicit cost figures.

**Baselines.** We restrict comparisons to methods that share FiPS's regime: post-training, factorization-based MLP compression without LoRA or distillation, so that differences reflect the factorization strategy rather than auxiliary training signals. We benchmark against three SVD-based LLM compression methods: ASVD (Yuan et al., 2023), which scales weights by activation statistics to mitigate outliers; SVD-LLM (Wang et al., 2024b), which employs truncation-aware whitening; and SVD-LLM V2 (Wang et al., 2025), which adds layer-wise rank allocation. All baseline metrics are from the original publications; we exclude LoRA (Hu et al., 2021) enhancements for fair comparison. Moreover, we compare FiPS against Basis Sharing (Wang et al., 2024a), which also pursues cross-layer parameter sharing but relies on dense coefficients. In contrast, FiPS enforces sparsity, which our ablations show to be essential (see Table 5).

**Evaluation.** We report perplexity on WikiText-2 (Merity et al., 2016) and C4 (Dodge et al., 2021), six classification benchmarks, and two generation tasks (TruthfulQA, GSM8K) via LM-Evaluation-Harness (Sutawika et al., 2023). At 20% compression (Table 3), FiPS achieves the lowest perplexity on both WikiText-2 and

| | Method | WikiText-2↓ | C4↓ | Openb. | ARC-e | WinoG. | HellaS. | PIQA | MathQA | **Avg.↑** | TruthfulQA↑ | GSM8K↑ |
|---|---|---|---|---|---|---|---|---|---|---|---|---|
| LLAMA-7B | Original | 5.68 | 7.34 | 0.34 | 0.75 | 0.70 | 0.57 | 0.79 | 0.27 | 0.57 | 0.30 | 0.09 |
| | ASVD | 11.14 | 15.93 | 0.29 | 0.53 | 0.64 | 0.41 | 0.68 | 0.17 | 0.45 | 0.21 | 0.04 |
| | SVD-LLM | 7.94 | 15.84 | 0.31 | 0.71 | 0.68 | 0.49 | 0.71 | 0.22 | 0.52 | 0.24 | 0.06 |
| | SVD-LLM V2 | 7.12 | 10.47 | 0.32 | 0.72 | 0.70 | 0.52 | 0.75 | 0.24 | 0.54 | **0.27** | **0.07** |
| | Basis Sharing | 7.74 | 15.03 | 0.28 | 0.66 | 0.66 | 0.46 | 0.71 | 0.25 | 0.50 | – | – |
| | FiPS (ours) | **6.06** | **8.10** | **0.32** | **0.72** | **0.70** | **0.56** | **0.78** | **0.26** | **0.56** | **0.27** | **0.07** |
| LLAMA-3.1-8B | Original | 6.14 | 9.47 | 0.35 | 0.80 | 0.73 | 0.60 | 0.80 | 0.40 | 0.61 | 0.49 | 0.45 |
| | ASVD | 17.55 | 28.41 | 0.20 | 0.59 | 0.61 | 0.41 | 0.69 | 0.30 | 0.47 | 0.37 | 0.28 |
| | SVD-LLM | 11.82 | 20.05 | 0.29 | 0.77 | 0.64 | 0.51 | 0.72 | 0.30 | 0.54 | 0.45 | 0.31 |
| | SVD-LLM V2 | 8.01 | 11.72 | **0.33** | **0.79** | 0.70 | 0.58 | 0.77 | 0.36 | 0.59 | **0.46** | 0.40 |
| | FiPS (ours) | **6.88** | **10.78** | **0.33** | **0.79** | **0.72** | **0.59** | **0.78** | **0.38** | **0.60** | **0.46** | **0.42** |

Table 3: **LLM Compression Results.** LLAMA-7B and LLAMA-3.1-8B at 20% compression. PPL↓: perplexity on WikiText-2 and C4; Avg.↑: mean over six classification tasks; TruthfulQA (BLEU↑) and GSM8K (exact match↑). ASVD from Yuan et al. (2023); SVD-LLM from Wang et al. (2024b); SVD-LLM V2 from Wang et al. (2025); Basis Sharing from Wang et al. (2024a). All results follow Sutawika et al. (2023); – indicates unreported metrics.

C4 for LLAMA-7B and LLAMA-3.1-8B, while matching or outperforming all baselines on downstream tasks. Results at 40% compression are provided in Table 6.

**Quantization-Aware Training (QAT).** To assess FiPS under low-precision regimes, we apply QAT to GEMMA-2-2B. § 4.2 reports WikiText-2 perplexity for 4-bit and 2-bit QAT (4× and 8× compression) as well as FiPS combined with 3-bit QAT. While 4-bit QAT matches `bfloat16` performance, 2-bit QAT degrades severely (PPL = 41.86). Combining 3-bit QAT with FiPS achieves the same 8× compression as 2-bit QAT but with substantially lower perplexity (35.43 vs. 41.86). Both approaches degrade substantially from the `bfloat16` baseline; the key finding is that FiPS recovers quality relative to aggressive quantization at the same compression factor.

| Variant | Prec. | Comp. | PPL ↓ |
|---|---|---|---|
| Baseline | BF16 | 1.0× | 15.61 |
| QAT | INT4 | 4.0× | 16.86 |
| QAT | INT2 | 8.0× | 41.86 |
| FiPS | BF16 | 1.5× | 32.01 |
| FiPS+QAT | INT3 | 8.0× | 35.43 |

Table 4: **QAT Results.** 3-bit QAT with FiPS achieves 8× compression at lower PPL than 2-bit QAT alone.

## 5 Ablations

We examine the importance of various components of the FiPS algorithm when compressing DEIT-B at 25% parameter budget. We ablate the following key components:

1. **Random Initialization (RI):** Using RI instead of SVD initialization results in a 1 percentage point drop in accuracy.
2. **Global Pruning (GP):** Using GP when sparsifying the factors **V** yields a 0.4 percentage point improvement over local pruning (LP), which enforces the same sparsity level for each group.
3. **Scaling Vectors (SV):** Following Liu et al. (2024), FC weights are normalized, and the magnitudes initialize the SV for neuron scaling. This enhances LP but is less effective than GP.

An analysis on sparsity, methods, and calibration settings using DEIT-B confirms that GMP with 75% sparsity and 20 epochs over 20 batches yields optimal performance (Figure 4). Figure 2a shows the rank–sparsity trade-off at a fixed 25% parameter budget: as sparsity on **V** increases from 0% to ∼80%, the rank $r$ grows to compensate, and reconstruction error decreases until diminishing returns set in beyond 80% sparsity. Table 5 compares all sparsification strategies across budgets for both DEIT-B and SWIN-L: GMP consistently achieves the highest accuracy, while the dense baseline (no sparsity on **V**) collapses at low budgets (e.g., 15.35% at 10%), underscoring that sparsity is not merely helpful but essential. Figure 4 further shows that 75% sparsity is optimal and that performance is robust to calibration data volume beyond 20 batches.

| Parameter Budget (Comp. Ratio) | 10% (∼57%) | | 25% (∼45%) | | 40% (∼33%) | | 50% (∼25%) | | 75% (∼6%) | |
|---|---|---|---|---|---|---|---|---|---|---|
| Method / Model | DEIT | SWIN | DEIT | SWIN | DEIT | SWIN | DEIT | SWIN | DEIT | SWIN |
| Dense | 15.35 | 3.61 | 65.71 | 60.31 | 74.33 | 80.61 | 79.22 | 83.59 | 81.36 | 85.64 |
| Static Sparsity | 65.26 | 65.6 | 80.06 | 84.37 | 81.48 | **85.69** | 81.70 | 85.98 | **81.86** | **86.23** |
| RigL | 66.67 | 70.96 | 80.31 | 84.57 | 81.50 | 85.59 | 81.65 | 85.91 | 81.82 | 86.20 |
| GMP (FiPS) | **70.04** | **74.04** | **80.64** | **84.78** | **81.69** | **85.69** | **81.83** | **85.99** | 81.82 | 86.21 |

Table 5: **Sparsification Method Comparison.** ImageNet top-1 validation accuracy (%) of DEIT-B (81.85%) (Touvron et al., 2021) and SWIN-L (86.24%) (Liu et al., 2021) compressed with FiPS using different sparsity methods. GMP consistently outperforms alternatives; the dense baseline collapses at lower budgets.

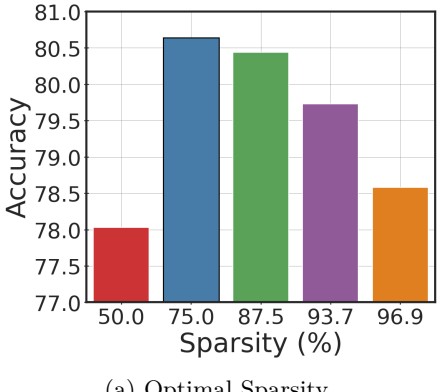

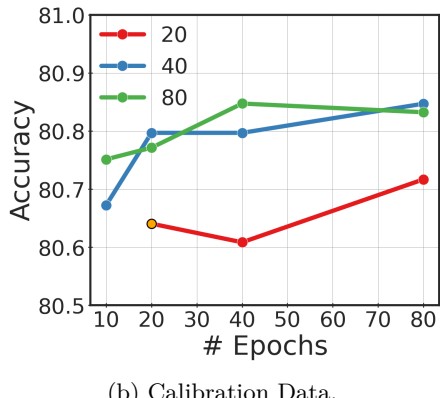

(a) Optimal Sparsity.

(b) Calibration Data.

Figure 4: **Sensitivity Analysis.** (a) Impact of sparsity levels on DEIT-B accuracy; 75% is optimal. (b) Effect of calibration data volume and training duration; 20 epochs over 20 batches suffices.

**Sparsity Distribution and MSE Loss.** Initial experiments in Figure 3a reveal that later layers incur higher reconstruction error under uniform compression budgets, suggesting they benefit from greater parameter allocation. FiPS addresses this by applying global magnitude pruning to its sparse factors, and indeed assigns more parameters to later layers (Figure 5b). Moreover, Figure 5c shows a strong negative correlation (−0.922) between the final sparsity pattern and the MSE losses in Figure 3a, confirming the effectiveness of this adaptive allocation.

# 6 Related Work

**Vision Transformers (ViT) & Large Language Models (LLM).** Recent transformer architectures extend beyond the foundational ViT models (Dosovitskiy et al., 2021), which treat image patches as token sequences. DEIT (Touvron et al., 2021) enhances data efficiency with distillation tokens, while SWIN (Liu et al., 2021) introduces a hierarchical design using shifted windows. Both vision models employ two FC layers per MLP module. In contrast, decoder-only LLMs such as LLAMA (Touvron et al., 2023), LLAMA-3 (Grattafiori et al., 2024), and GEMMA-2 (Team et al., 2024) achieve state-of-the-art zero-shot and instruction-following performance using three FC layers per MLP module in each block.

**Sparsity in Neural Networks.** Early methods involved heuristic pruning, such as removing the smallest-magnitude parameters (Thimm & Fiesler, 1995). Later approaches, such as GMP (Zhu & Gupta, 2017), increased the extent of pruning, while dynamic pruning with accelerated schedulers was explored by Kurtic et al. (2023). Static sparsity uses a preinitialized mask throughout training (Hoefler et al., 2021), whereas dynamic methods such as RigL (Evci et al., 2021) adjust the sparsity pattern during training based on gradient information.

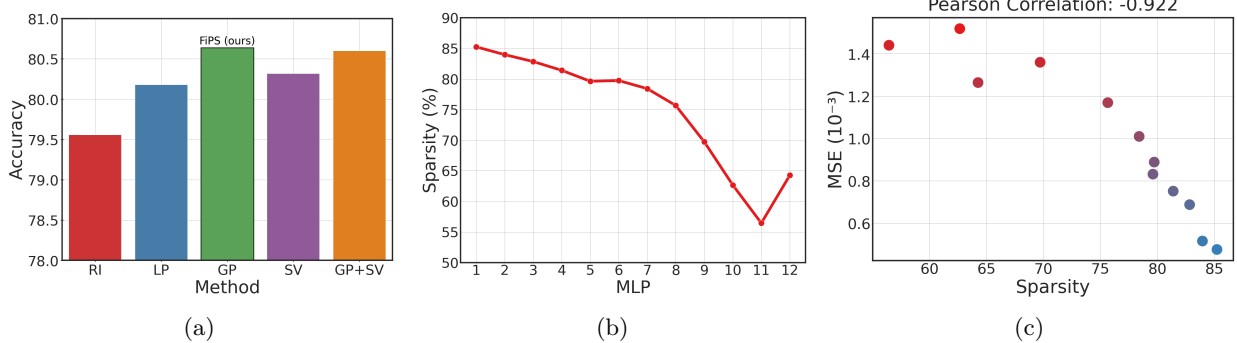

Figure 5: **DeiT-B Ablation and Global Sparsity Analysis.** (a) Component analysis of the FiPS algorithm: Random Initialization (RI), Local Pruning (LP), Global Pruning (GP), and Scaling Vectors (SV). (b) End-of-training sparsity allocation; later layers require more parameters. (c) Strong correlation between the MSE reported in Figure 3a and the parameter distribution captured by FiPS.

**Tensor Decomposition.** Yu & Wu (2023) introduce Adaptive Atomic Feature Mimicking (AAFM) and its global variant Global Feature Mimicking (GFM), which apply truncated PCA on ViT activations followed by fine-tuning. In LLMs, Activation-aware SVD (ASVD) (Yuan et al., 2023), truncation-aware SVD-LLM (Wang et al., 2024b), and SVD-LLM V2 with rank distillation and layer-wise allocation (Wang et al., 2025) serve as our baselines.

**Parameter Sharing.** Several prior works explore parameter sharing in neural networks. Eban et al. (2019) introduce a Sum-Product reducer to map shared parameters, and Obukhov et al. (2021) employ TR decomposition for shared parameters in 3D tensors. At the block level, ALBERT (Lan et al., 2020) shares entire transformer blocks (all parameters across layers), reducing model size but constraining every layer to identical weights. Zhang et al. (2022) propose "Weight Multiplexing," sharing parameters between MLP modules in ViTs with distillation and learned linear projections between blocks to aid recovery; however, MiniViT does not impose sparsity on its projection factors and relies on distillation for quality. Concurrent work by Wang et al. (2024a) also explores cross-layer parameter sharing via SVD, representing weight matrices as linear combinations of shared basis vectors with dense, layer-specific coefficients.

In summary, FiPS is distinguished from all the above compression approaches by combining three properties that no prior method jointly offers: (i) cross-block weight sharing via a shared basis $\mathbf{U}$, (ii) low-rank factorization initialized by SVD, and (iii) sparsity on the projection matrices $\mathbf{V}_i$, optimized via block-wise reconstruction loss. In particular, we show in § 2.1 that enforcing sparsity in $\mathbf{V}$ is critical for achieving lower reconstruction error at the same parameter budget compared to dense coefficients (Wang et al., 2024a). Methods such as pruning combined with distillation (Kurtic et al., 2023) and LoRA-based compression (Hu et al., 2021) operate in a complementary regime—they rely on auxiliary supervision signals or learned adapters—and are orthogonal to FiPS, as we demonstrate with QAT in § 4.2.

## 7 Conclusion

We presented FiPS, a framework for compressing transformer MLPs via fine-grained inter-layer parameter sharing that unifies cross-block weight tying, low-rank factorization, and sparsity in a single optimization. FiPS achieves state-of-the-art compression–accuracy trade-offs on MLP layers: up to 33% compression on ViTs with <1% accuracy loss (up to 57% with fine-tuning), up to 20% on LLMs while outperforming existing SVD-based methods at matched compression, and 8× on GEMMA-2-2B when combined with QAT. These findings establish parameter sharing as a competitive alternative to existing compression strategies. Future work includes extending to attention layers—whose projection matrices share the same block-repeated structure that FiPS exploits—quantizing the shared bases, and developing specialized kernels that keep $\mathbf{U}$ resident in fast memory for efficient on-device inference.

## Broader Impact Statement

This paper advances Machine Learning by introducing Fine-grained Parameter Sharing (FiPS), a model compression method that improves the efficiency of Vision Transformers (ViTs) and Large Language Models (LLMs). By leveraging parameter sharing, low-rank factorization, and sparsity, FiPS reduces computational and memory costs, enhancing AI accessibility on resource-constrained devices. While model compression promotes efficiency and sustainability, it may also enable broader AI deployment in sensitive domains with ethical concerns such as bias, misinformation, and privacy. Nevertheless, this work should not introduce new risks beyond those inherent in deep learning, but we encourage responsible deployment and ethical considerations in practice.

## Author Contributions

Cem Üyük led the project, proposed and executed the experimental plan, facilitated the team meetings, developed the software architecture, implemented static sparse training and provided code review for the sparse training algorithms, wrote the initial draft of the paper, and further contributed to writing significantly while also creating most of the plots. Mike Lasby implemented sparse training algorithms, assisted the software architecture development, handled distributed training integration, performed code reviews, and assisted with writing and proofreading the paper. Mohamed Yassin assisted with coding and running inference experiments. Utku Evci proposed the project and its central idea, contributed to the research plan and direction, advised Cem, reviewed the code, helped substantially with the writing, and created some of the plots. Yani Ioannou helped with the research direction, contributed to the paper's motivation, helped with the writing, provided compute resources, and supervised the work by members of the Calgary ML Lab at the University of Calgary, including Cem Üyük (Visiting Student Researcher), Mike Lasby (PhD Student), and Mohamed Yassin (Research Assistant).

## Acknowledgments

We gratefully acknowledge the support of Alberta Innovates (ALLRP 577350-22, ALLRP 600038-24), the Natural Sciences and Engineering Research Council of Canada (NSERC) (RGPIN-2022-03120, DGECR-2022-00358), Defence Research and Development Canada (DGDND-2022-03120), and NSERC/Agence Nationale de la Recherche (ANR) (ALLRP 602719-24). This project was undertaken thanks to funding from IVADO and the Canada First Research Excellence Fund. This research was enabled in part by support provided by the Digital Research Alliance of Canada (alliancecan.ca) and Google Cloud. We also acknowledge Erik Schultheis' very helpful feedback with regard to custom kernel design.

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

# A   Appendix

## A.1   Method

Referring to § 2.2, we detail four methods for concatenating the FC weights of two MLPs (four matrices $\mathbf{W}_{ij} \in \mathbb{R}^{d \times p}$, with $p = 4d$ and $i, j \in \{1, 2\}$):

I. **Full long-axis concatenation**, forming

$$\mathbf{W} = [\,\mathbf{W}_{11}\,\mathbf{W}_{12}\,\mathbf{W}_{21}\,\mathbf{W}_{22}\,] \ \in \ \mathbb{R}^{d \times 16d}.$$

II. **Module-wise long + inter-module short:**

$$\mathbf{W}_A = [\mathbf{W}_{11}\mathbf{W}_{12}] \in \mathbb{R}^{d \times 8d}$$

$$\mathbf{W}_B = [\mathbf{W}_{21}\mathbf{W}_{22}] \in \mathbb{R}^{d \times 8d}$$

$$\mathbf{W} = \begin{bmatrix} \mathbf{W}_A \\ \mathbf{W}_B \end{bmatrix} \ \in \ \mathbb{R}^{2d \times 8d}.$$

III. **Module-wise short + inter-module long:**

$$\mathbf{W}_C = [\mathbf{W}_{11}\mathbf{W}_{21}] \in \mathbb{R}^{2d \times 4d}$$

$$\mathbf{W}_D = [\mathbf{W}_{12}\mathbf{W}_{22}] \in \mathbb{R}^{2d \times 4d}$$

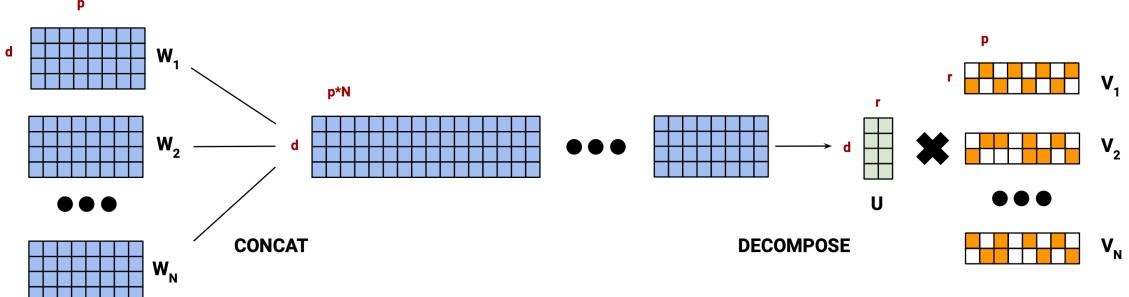

Figure 6: **Parameter Sharing Through Sparse Tensor Decomposition.** A group of FC layers are concatenated along the larger dimension, $p$, and decomposed into two matrices: a shared basis, $\mathbf{U}$, and a sparse projection matrix, which is then sliced up respectively for each layer.

,

$$\mathbf{W} = [\, \mathbf{W}_C \, \mathbf{W}_D \,] \ \in \ \mathbb{R}^{2d \times 8d}.$$

IV. **Full short-axis concatenation**, yielding

$$\mathbf{W} = \begin{bmatrix} \mathbf{W}_{11} \\ \mathbf{W}_{12} \\ \mathbf{W}_{21} \\ \mathbf{W}_{22} \end{bmatrix} \ \in \ \mathbb{R}^{4d \times 4d}.$$

These configurations are evaluated in Figure 2b and discussed in § 2.2.

The overall method is depicted in Figure 6, and further details are explained below.

### A.1.1 Growing Neurons in Shared Bases and Sparse Factors

As discussed in § 3, high parameter budgets and sparsity levels (e.g., 26.5% parameter budget, 75% sparsity, and groups of four blocks in DEIT-B) often result in the rank $r$ exceeding the model dimension $d$. Since SVD yields only $d$ initialization directions, we investigate three methods to initialize the remaining $k = r - d$ dimensions:

1. **Random Growth:** Initialize new neurons in $\mathbf{U}$ to zero and in $\mathbf{V}$ randomly using He et al. (2015);

2. **Neuron Splitting:** Duplicate the top $k$ neurons of $\mathbf{U}$ and halve the top $k$ neurons of $\mathbf{V}$, following Chen et al. (2016);

3. **Hybrid Initialization:** Initialize new neurons in $\mathbf{U}$ to zero and derive those in $\mathbf{V}$ from the top $k$ neurons, scaled by $\tau$. This minimizes the immediate impact of new neurons in $\mathbf{V}$, allowing their gradual reactivation, as proposed by Evci et al. (2022).

After performing a hyperparameter sweep for $\tau$, hybrid initialization outperformed the alternatives, achieving 1% and 2% higher accuracy than methods (1) and (2), respectively.

### A.2 Compression Ratio Computation

The reported compression ratio accounts for all storage overhead, including sparse metadata and uncompressed components:

- **Sparse factors ($\mathbf{V}$):** We store both the nonzero values and their positions. For unstructured sparsity, we use a bitmap mask (1 bit per element). For 2:4 structured sparsity, the fixed pattern eliminates the need for explicit indices.

- **Shared basis (U):** Each group has one shared basis **U**, counted once per group. For example, with $\beta = [4, 6, 6, 6, 6, 4]$ for LLAMA-7B (6 groups), we store 6 shared bases.

- **Non-MLP parameters:** Attention projections, embeddings, and layer norms remain uncompressed and are counted at full precision.

- **Quantized setup:** Each parameter is counted at its precision level (e.g., 3 bits for INT3), and masks are counted as 1 bit per element.

Formally, the compression ratio is:

$$\text{Compression Ratio} = \frac{|\theta_{\text{non-MLP}}| \cdot b + G \cdot |\mathbf{U}| \cdot b + \sum_i \left[(1 - s) \cdot |\mathbf{V}_i| \cdot b + |\mathbf{V}_i| \cdot 1\right]}{|\theta_{\text{original}}| \cdot b_{\text{original}}}, \tag{4}$$

where $G$ is the number of groups, $s$ is the sparsity level, $b$ is the bits per parameter (e.g., 16 for `bfloat16`), $b_{\text{original}}$ is the original precision, and the sum is over all sparse factors $\mathbf{V}_i$ across all layers. For 2:4 structured sparsity, the mask term $|\mathbf{V}_i| \cdot 1$ is omitted as the sparsity pattern is implicit.

### A.3 Model Links

- DEIT-B (Touvron et al., 2021): https://huggingface.co/facebook/deit-base-patch16-224

- SWIN-L (Liu et al., 2021): https://huggingface.co/microsoft/swin-large-patch4-window7-224

- GEMMA-2-2B-IT (Team et al., 2024): https://huggingface.co/google/gemma-2-2b-it

- GEMMA-2-9B (Team et al., 2024): https://huggingface.co/google/gemma-2-9b

- LLAMA-7B (Touvron et al., 2023): https://huggingface.co/huggyllama/llama-7b

- LLAMA-3.1-8B (Grattafiori et al., 2024): https://huggingface.co/meta-Llama/Llama-3.1-8B

### A.4 Further LLM Results

| Variant | Comp. Ratio | PPL ↓ |
|---|---|---|
| Original | 0% | 7.34 |
| SVD-LLM | 20% | 15.84 |
| SVD-LLM V2 | 20% | 11.72 |
| FiPS (ours) | 20% | **8.10** |
| SVD-LLM | 40% | 75.42 |
| FiPS (ours) | 40% | **10.57** |

Table 6: Perplexity on C4 for Llama7B under different compression ratios applied with FiPS and the baselines.

### A.5 Different Sparsification Methods

The sparsification method comparison is presented in Table 5 in the main text (§5). For DEIT-B, *RigL* consistently outperforms both *Dense* and *Static Sparsity* across parameter budgets ranging from 10% to 50%. At higher parameter budgets, all methods converge to similar accuracies approaching the original model's performance. For SWIN-L, *RigL* surpasses *Dense* and *Static Sparsity* at 10% and 25% parameter budgets. However, at higher parameter budgets, *Static Sparsity* achieves slightly higher accuracies.

| Comp. Ratio | 10% | 25% | 40% | 50% | 75% |
|---|---|---|---|---|---|
| STE | 42.89 | 73.26 | 78.26 | 79.36 | 78.89 |
| SR-STE | 45.31 | 75.53 | 79.71 | 80.68 | 81.24 |
| NMSRigL | 44.87 | 75.71 | 79.97 | 80.99 | 81.40 |
| NMSGMP | 52.36 | 76.88 | 80.59 | 81.31 | 81.51 |
| FiPS (50% Sparsity) | 54.00 | 77.56 | 80.94 | 81.63 | 81.77 |
| FiPS (75% Sparsity) | 70.04 | 80.64 | 81.69 | 81.83 | 81.82 |

Table 7: **Structured Sparsity Performance.** ImageNet top-1 accuracy (%) of DEIT-B (81.85%) (Touvron et al., 2021) for various structured sparsification methods at 50% and 75% sparsity, compared to unstructured FiPS. Methods include Straight Through Estimator (*STE*), Sparse-Refined STE, N:M Structured RigL (*NMSRigL*), and N:M Structured GMP (*NMSGMP*) at 50% sparsity, corresponding to 2:4 structures (Lee et al., 2023; Zhou et al., 2021; Lasby et al., 2024).

## A.6 Structured Sparsity

We evaluate the generalization performance of FiPS using structured sparsity, with results presented in Table 7. The methods evaluated include the Straight Through Estimator (*STE*), which employs top-$k$ weight magnitude selection, projects parameters into a sparse subspace during training, and applies gradients to dense parameters through a gradual pruning schedule; Sparse-Refined STE (*SR-STE*) (Zhou et al., 2021), which mitigates the adverse effects of approximated gradients; and N:M Structured RigL (*NMSRigL*) and N:M Structured GMP (*NMSGMP*) (Lee et al., 2023; Lasby et al., 2024), where N:M specifies the sparsity pattern of the weight matrix (e.g., 50% sparsity in FC matrices of size $d \times 4d$ corresponds to a 2:4 structure).

## A.7 Sensitivity Analysis

The sensitivity analysis plots are presented in Figure 4 in the main text (§5).

**Calibration Dataset Size and Training Length.** We examine how the number of calibration batches and training epochs affects performance using a fixed batch size of 128. To ensure at least one example from each category, we begin with a minimum of 10 batches and also evaluate 20, 40, and 80 batches. After filtering out configurations more than 0.25% below the highest accuracy, we adopt the most efficient setting of 20 epochs over 20 batches for all reported results, as shown in Figure 4b.

**Optimal Sparsity for Sparse Factors.** We compressed DEIT-B as described in § 4.1, using sparsity levels ranging from 50% to 96.9% (Figure 4a). The best performance was observed at 75% sparsity. While increasing sparsity to 87% yielded similar accuracy, lowering it to 50% resulted in a notable performance drop, likely due to a significant reduction in rank.

## A.8 Hyper-parameters

### A.8.1 Ablation on the Block–Grouping Hyper-parameter $\beta$

**Definition.** We define $\beta$ as an *ordered list* whose $i^{\text{th}}$ element gives the number of consecutive decoder blocks whose MLP weights are tied in the $i^{\text{th}}$ parameter sharing group:

$$\beta = \big[\beta_1,\ \beta_2,\ \ldots,\ \beta_G\big], \quad \sum_{g=1}^{G} \beta_g = L,$$

where $L$ is the total number of decoder blocks. Self-attention parameters remain untied in all experiments. For every architecture we sweep over a small number of plausible $\beta$ lists (3–5 candidates) and keep the one with the lowest validation perplexity (PPL) after compression.

Table 8: **Block–grouping sweep for Gemma-2-2B.** Each row lists the candidate $\beta$ and the resulting validation perplexity (PPL) on WikiText-2. Baseline (no sharing) PPL is 15.61; lower is better.

| Config. | $\beta$ list | PPL $\downarrow$ |
|---|---|---|
| 1 | $[\,4, 4, 5, 5, 4, 4\,]$ | 21.42 |
| 2 | $[\,1, 4, 4, 4, 4, 4, 4, 1\,]$ | $\approx$23 |
| 3 | $[\,2, 3, 4, 4, 4, 4, 3, 2\,]$ | $\approx$22 |
| 4 | $[\,2, 6, 6, 6, 6\,]$ | 21.81 |
| 5 | $[\,3, 5, 5, 5, 5, 3\,]$ | 21.86 |

**Default Heuristic.** Our analysis in § 2.3 shows that adjacent layers exhibit smaller reconstruction error when sharing, and deeper layers require more capacity. This motivates a "tapered" grouping where group sizes are smaller at the network extremes and larger in the middle. Based on our experiments, we propose the following reproducible default for a decoder with $L$ blocks:

> Use 4–6 groups with $\beta$ tapered as $[small, medium, \dots, medium, small]$, where $small \approx L/8$ and $medium \approx L/5$.

For example:

- $L = 32$ (Llama): $\beta = [4, 6, 6, 6, 6, 4]$

- $L = 26$ (Gemma-2-2B): $\beta = [4, 4, 5, 5, 4, 4]$

- $L = 12$ (DeiT-B): $\beta = [4, 4, 4]$

**Block Groups of ViTs.** Using the list-valued notation for $\beta$ introduced above, we set

$$\beta_{\text{DeiT-B}} = [4,\, 4,\, 4],$$

i.e. three groups of four consecutive blocks (each block contains one MLP).

The depth pattern of Swin-L is 2+2+18+2 blocks. We tie MLP weights inside every 2-block stage and split the 18-block stage into three groups of six, which gives

$$\beta_{\text{Swin-L}} = [2,\, 2,\, 6,\, 6,\, 6,\, 2].$$

**Gemma-2-2B.** Table 8 shows the five $\beta$ lists evaluated for Gemma-2-2B-IT at 20% compression. Config. 1—$\beta = [\,4, 4, 5, 5, 4, 4\,]$—yields the lowest PPL and is therefore used in the main paper. Notably, alternative groupings yield PPL within $\sim$2 points of the optimal (21.42 vs. $\approx$23), indicating that FiPS is not overly sensitive to this hyperparameter.

**Llama-2-7B.** With $L = 32$ decoder blocks we compared three $\beta$ lists:

| Candidate $\beta$ | Block groups (sizes) | Observation |
|---|---|---|
| $[\,4, 4, 4, 4, 4, 4, 4, 4\,]$ | 8 groups $\times$ 4 blocks | Highest PPL |
| $[\,\mathbf{4, 6, 6, 6, 6, 4}\,]$ | **6 groups with sizes** $(4, 6, 6, 6, 6, 4)$ | **Best PPL; used in §4.1** |
| $[\,8, 8, 8, 8\,]$ | 4 groups $\times$ 8 blocks | Slightly worse than above |

A gently varying list—with smaller groups at the extremes and larger groups in the middle—provides the best compression/accuracy trade-off.

**Llama-3.1-8B.** The 8B model shares the same 32-layer decoder. We reused the winning $\beta$ from the 7B sweep,

$$\beta_{\text{opt}} = [\,4, 6, 6, 6, 6, 4\,],$$

because (i) it keeps the total tied-parameter ratio identical and (ii) in a spot-check it preserved PPL within $+3.5$ of the uncompressed baseline—better than the uniform alternatives $[\,8, 8, 8, 8\,]$ or $[\,4, 4, 4, 4, 4, 4, 4, 4\,]$. This demonstrates that the optimal $\beta$ transfers across models of similar architecture.

**Key Insights.**

- Optimal $\beta$ often starts and ends with smaller groups, reflecting the intuition that early and late layers contain more specialized features.

- Extremely fine-grained sharing (e.g., many 4-block groups) hurts accuracy, while overly coarse sharing (uniform 8-block groups) sacrifices capacity.

- For Llama models, a tapered list such as $[4, 6, 6, 6, 6, 4]$ ties roughly 22–25% of MLP parameters yet adds only $\sim$3–4 perplexity points.

- The method is robust to the choice of $\beta$: alternative groupings typically yield performance within 2 PPL points of the optimal.

These ablations inform all main-text compression results.

### A.8.2  Optimizer

**ViT Compression.** To minimize local error, we employ a logarithmic grid for hyper-parameter tuning. The learning rates for Dense, Static Sparsity, GMP, and RigL are set as follows for both DEIT-B and SWIN-L:

1. Dense: $1.25 \times 10^{-4}$,

2. Static Sparsity: $2.5 \times 10^{-4}$,

3. GMP: $1 \times 10^{-3}$,

4. RigL: $1 \times 10^{-3}$.

**ViT Transfer Learning.** We use a linear grid, as some hyper-parameters are derived from the codebase of DEIT. The optimal learning rates for FiPS are:

1. CIFAR-100: $2.5 \times 10^{-5}$;

2. Flowers102: $1 \times 10^{-4}$;

3. Oxford-III-Pets: $7.5 \times 10^{-6}$;

4. iNaturalist 2019: $1 \times 10^{-4}$.

**LLMs.** Eight logarithmically spaced values were swept. The final values for FiPS are:

1. GEMMA-2-2B: $4.0 \times 10^{-4}$,

2. LLAMA-7B: $1.0 \times 10^{-6}$,

3. LLAMA-3.1-8B: $1.0 \times 10^{-5}$.

**Sparsifier**

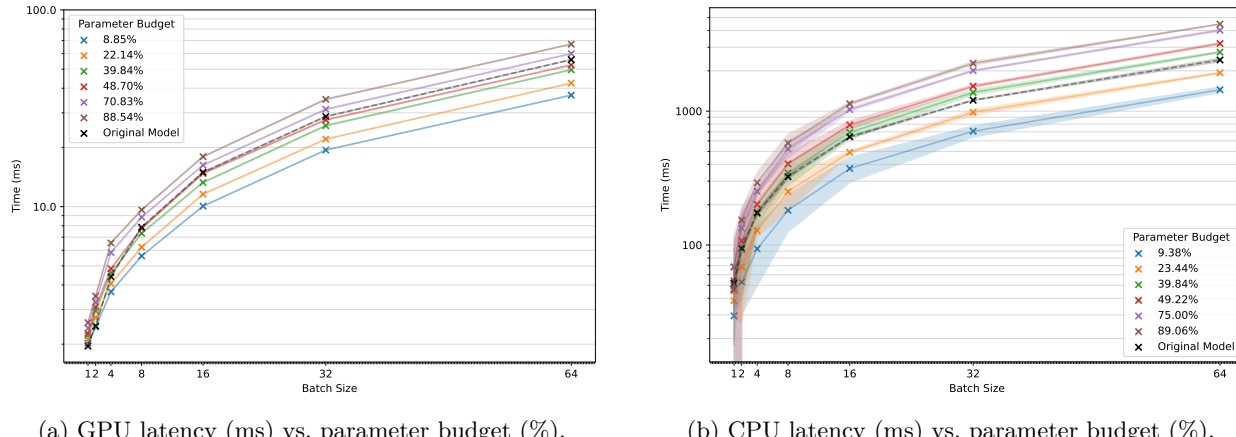

(a) GPU latency (ms) vs. parameter budget (%).      (b) CPU latency (ms) vs. parameter budget (%).

Figure 7: **DeiT-B inference latency benchmarks.** (a) End-to-end latency of 2:4 sparse FiPS on an NVIDIA A4000 for batch sizes 1–64; a 22 % parameter budget yields a 25 % speed-up once the batch size exceeds 8. (b) Latency of 75 % unstructured sparse FiPS accelerated by DeepSparse on an Intel Xeon W-2145 CPU, outperforming the dense model at every tested batch size. Shaded regions denote $\pm 1\sigma$ over runs collected by `torch.benchmark.Timer.blocked_autorange`; GPU timings exclude `torch.compile` warmup iterations.

**Gradual Magnitude Pruning (GMP).** GMP begins with an initial sparsity level of 25%. During training, sparsity is gradually increased to 50% at the 25% training mark and ultimately reaches 75% by training completion. We use $\Delta T = 50$ for update steps.

**RigL.** RigL employs an initialization phase that combines pruning with a growth ratio of 0.1 for block-wise error minimization and a growth ratio of 0.05 for transfer learning tasks, with $\Delta T = 50$ for growth and pruning steps. This conservative growth ratio in transfer learning helps preserve the mask obtained during initial training, ensuring that learned masks are retained.

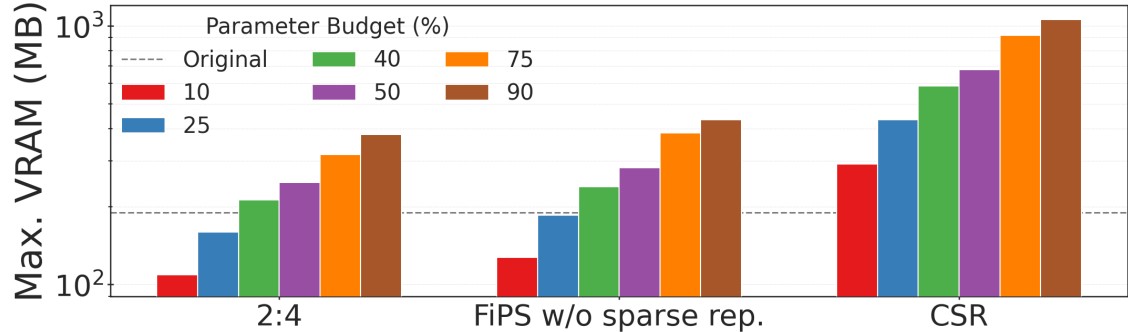

Figure 8: **DeiT-B Inference Memory Profile.** Maximum VRAM allocation at batch size 64 for 50% sparse FiPS using 2:4, strided (dense), and CSR tensor formats. At 10% and 25% parameter budgets, 2:4 sparsity reduces peak memory by 44% and 18%, respectively; CSR incurs higher overhead at modest sparsity due to index storage.

### A.9   Latency and Memory Profiling

As discussed in § 3, high levels of sparsity and parameter budgets can result in the SVD rank exceeding a model's hidden dimension. For instance, in DEIT-B, achieving 75% sparsity under parameter budget constraints exceeding 26.5% with four block groups increases the rank of the shared singular vectors beyond the original model's embedding dimension. Efficient sparse operations and representations are crucial for minimizing the latency and memory overhead introduced by FiPS. Figure 7 and Figure 8 summarize the

latency and memory results for DEIT-B compressed with FiPS using 2:4 structured GMP, highlighting the resulting speedups and memory savings on both CPU and GPU platforms.

