# OpenReview forum: "Learning Fine-grained Parameter Sharing via Sparse Tensor Decomposition"
_TMLR — Accepted by TMLR_

### Review · Reviewer_6GP7 · 2026-03-20

**Summary Of Contributions:**

The paper introduces a novel framework for compressing transformer models by combining low-rank tensor decomposition, cross-layer parameter sharing, and sparsity. The idea is to decompose concatenated MLP weight matrices across layers into a shared low-rank basis and layer-specific sparse projection matrices, enabling sharing at the neuron level instead of block/layer level.

Strengths
- Considering neuron level is a novel approach in parameter sharing
- Motivated design choices and good ablation study (e.g., sparsity on V, grouping strategies).
- promising results in the experimental section
- con be used together quantization, increasing practical utility.

Weaknesses
- lacks of formality in the theoretical parts, the method is largely heuristic (e.g., grouping strategy, sparsity schedules) without formal guarantees.
- The evaluations almost only consider MLP layers while attention layers are not considered.
- lacks of comparison to broader compression techniques (e.g., pruning+distillation, LoRA-based compression).
- training cost and complexity (multi-stage optimization, calibration dataset) are non-trivial and not sifficiently analyzed.

**Audience:**

Yes

**Audience Explanation:**

I think TMLR's audience may be interested in the paper as it considers efficient compression of large models with a novel fine-grained parameter sharing approach and significant empirical results.

**Broader Impact Concerns:**

No ethical concerns

**Claims And Evidence:**

Yes

**Claims Explanation:**

Overall, the claims are mostly supported, but with some issues, as also reported in the weaknesses.
Basically, the empirical evidence is strong but not exhaustive. The claims are largely credible but somewhat overgeneralized relative to the experimental scope.

For example:
- no variance reporting, confidence intervals, or multiple runs, so results may not reflect stability.
- limited comparison with other approaches, sucha s distillation-based compression or structured pruning.
- missing details on computational costs, especially for training and calibration overhead

So, in general, the main claim about "unified compression framework" is overstated since attention layers and other architectures are not evaluated.

More in detail:
- the latency and memory claims are, in my opinion, not fully supported. The paper states that FiPS improves both memory efficiency and computational speed, showing the results in Figure 7 and Figure 8. Figures support speedups and memory reductions on the tested hardware, but in general these experiments are quite narrow: one model family, specific sparsity format, specific hardware/software stacks, and only inference. Also, Figure 7 seems to report latency curves without statistical variability that can help ensuring that in the considered scenario the prposed appoach effectively performs better. Also considering other results, the abstract claims that FiPS compresses models by 20-50% with negligible quality degradation. This is, in my opinion, only partially supported. On ViTs, yes, quality loss is very small at some operating points. But for LLMs the evidence is more mixed: at 20% compression, Llama-7B WikiText-2 perplexity increases from 5.68 to 6.06 and C4 from 7.34 to 8.10, which is modest; however, the appendix's Table 5 shows that at 40% compression on Llama-7B, C4 perplexity rises to 10.57 from 7.34, which is a substantial degradation. So "20–50% with negligible degradation" overgeneralizes. Also here statistical analysis could be added to demonstrate that FiPS outperforms competitors as in some cases the differences in results are not so high (e.g., FipS vs SVD-LLM V2 in Table 3).
- the claim that FiPS is a "unified compression framework" for transformers is only partially supported because the method is evaluated only on MLP layers, not attention layers. However, I acknowledge the authors to have openly stated that the paper focuses only on MLP and even lists extending to attention as future work in the conclusion. But for these motivations, the paper should better frame what it means with unified.
- the claim that FiPS "consistently achieves the highest accuracy with the lowest overhead" in the ImageNet discussion is also only partly supported. Accuracy is supported by Table 1 for the shown baselines, but "lowest overhead" is not demonstrated with comparable quantitative overhead measurements against AAFM/GFM in the same table. The text says FiPS and FiPS+FT match AAFM/GFM in compute and memory budgets, but detailed compute-cost comparisons are not reported there. So the accuracy part is evidenced while the overhead part is asserted more than demonstrated.

**Requested Changes:**

I suggest the following changes:
- improve the baseline comparisons by including for example pruning + distillation pipelines or quantization only methods.
- improve the statistical validation of the results by computing mean and standard dev on multiple runs, considering multiple seeds.
- add more formal details on grouping strategy beta, neuron growth mechanism, tau scaling, and about the choice of value r considering parameters budget.
- add information about training costs.

As regards, this last point, it could be usefull to add calibration dataset size vs performance trade-off, running time compared with competitors, and/or memory footprints.

It would be also a strong addition considering other parts of the networks, such as attention layers (key limitation acknowledged but not tested), or completely different architectures such as anocder-decoder.
Otherwise I soul suggest rewriting some claims to consider MLP compression only.

Also, in the ablation section, some parts could be added. For example, trade-off between rank and sparsity, impact of removing each component (e.g., no sharing, no sparsity).

Finally, while it is possible to use appendices, a paper should be self-contained also without them. So, I would suggest reducing the use of appendices for key implementation details when possible.

---

> ### Comment · Action_Editor_Eq96 · 2026-03-20
> **Please update your review**
>
> Dear Reviewer 6GP7,
>
> Thank you for submitting the review in time
>
> Could you please clarify
> 1. what claims made in the submission are (or NOT) supported by accurate, convincing and clear evidence and
> 2. what the evidence is
>
> ?
>
> Thank you,
> Your AE

---

> ### Comment · Reviewer_6GP7 · 2026-03-23
> **About claims**
>
> Dear AE, dear authors,
> sorry for having under-explained my cocnerns. I hope the following will fully explain them.
>
> The latency and memory claims are, in my opinion, not fully supported. The paper states that FiPS improves both memory efficiency and computational speed, showing the results in Figure 7 and Figure 8. Figures support speedups and memory reductions on the tested hardware, but in general these experiments are quite narrow: one model family, specific sparsity format, specific hardware/software stacks, and only inference. Also, Figure 7 seems to report latency curves without statistical variability that can help ensuring that in the considered scenario the prposed appoach effectively performs better.
> Also considering other results, the abstract claims that FiPS compresses models by 20-50% with negligible quality degradation. This is, in my opinion, only partially supported. On ViTs, yes, quality loss is very small at some operating points. But for LLMs the evidence is more mixed: at 20% compression, Llama-7B WikiText-2 perplexity increases from 5.68 to 6.06 and C4 from 7.34 to 8.10, which is modest; however, the appendix's Table 5 shows that at 40% compression on Llama-7B, C4 perplexity rises to 10.57 from 7.34, which is a substantial degradation. So "20–50% with negligible degradation" overgeneralizes. Also here statistical analysis could be added to demonstrate that FiPS outperforms competitors as in some cases the differences in results are not so high (e.g., FipS vs SVD-LLM V2 in Table 3).
>
> The claim that FiPS is a "unified compression framework" for transformers is only partially supported because the method is evaluated only on MLP layers, not attention layers. However, I acknowledge the authors to have openly stated that the paper focuses only on MLP and even lists extending to attention as future work in the conclusion. But for these motivations, the paper should better frame what it means with unified.
>
> The claim that FiPS "consistently achieves the highest accuracy with the lowest overhead" in the ImageNet discussion is also only partly supported. Accuracy is supported by Table 1 for the shown baselines, but "lowest overhead" is not demonstrated with comparable quantitative overhead measurements against AAFM/GFM in the same table. The text says FiPS and FiPS+FT match AAFM/GFM in compute and memory budgets, but detailed compute-cost comparisons are not reported there. So the accuracy part is evidenced while the overhead part is asserted more than demonstrated.

---

> > ### Comment · Action_Editor_Eq96 · 2026-03-23
> >
> > Dear Reviewer 6GP7,
> >
> > Thank you for clarifying those.
> >
> > Could you please use "Edit" button attached to your review and update the corresponding section(s) so that other reviewers can see the comments when your review is released to them?
> >
> > Best,
> > Your AE

---

> ### Author Response · Authors · 2026-04-13
> **Response to Reviewer 6GP7 (1/2)**
>
> We thank the reviewer for their thorough evaluation and actionable suggestions. Below addresses the discussion points and the requested changes.
>
> ### Discussion
>
> **1. Latency/memory claims are narrow; "20–50% with negligible degradation" overgeneralizes.**
> The latency benchmarks do only cover DeiT-B on specific hardware, and we should have been upfront about that. We added a FLOP analysis in §4.1 to show the cost reduction generalizes in principle, while making clear that the measurements themselves are for DeiT-B. On "negligible degradation": for ViTs the quality loss really is under 1% top-1 at most operating points, but for LLMs the picture is more nuanced. We rewrote the abstract to report accurate compression ratios: "up to 33% compression with <1% accuracy loss (up to 57% with fine-tuning)" for ViTs and "up to 20% while outperforming existing SVD-based methods" for LLMs. Table 1 now includes a dedicated row mapping each parameter budget to its whole-model compression ratio.
>
> **2. "Unified compression framework" only partially supported.**
> We now say "a unified framework for compressing transformer MLPs" in the abstract. The word "unified" refers to combining sharing + factorization + sparsity, not to covering attention or other architectures.
>
> **3. "Lowest overhead" not demonstrated.**
> We replaced this with a qualitative comparison in §4.1: AAFM needs no fine-tuning, GFM needs full fine-tuning, and FiPS sits in between (cheaper than GFM, comparable to AAFM). We no longer claim "lowest."
>
> ### Requested Changes
>
> **1. Include pruning+distillation or quantization baselines.**
> FiPS is a post-training, layer-wise compression method based on matrix factorization. Our baselines are chosen to match this regime so that performance differences reflect the compression strategy rather than auxiliary training signals. For ViTs, AAFM performs layer-wise PCA-based compression with MSE error correction (directly analogous to FiPS), and GFM extends it with full fine-tuning (analogous to FiPS+FT). For LLMs, ASVD, SVD-LLM, SVD-LLM V2, and Basis Sharing are all recent, strong layer-wise factorization and sharing baselines from top-tier venues. Including pruning+distillation or LoRA-based methods would conflate the compression strategy with the benefit of auxiliary supervision or learned adapters.
>
> That said, the paper already contains evidence that sparsity (pruning) alone is suboptimal: Figure 2(a) shows that applying sparsity without sharing yields higher reconstruction error, and Figure 3(b) shows that the dense (no sparsity) variant collapses as group size grows. These are directionally informative for what layer-wise pruning alone would achieve. We also demonstrate orthogonality with quantization via QAT in Table 4.
>
> We are open to including additional baselines if the reviewer can suggest a specific method that operates in the same post-training, layer-wise regime without auxiliary supervision and has publicly available code for the models and compression ratios we evaluate. We would do our best to accommodate this within the revision timeline.
>
> *Changes: §4.1, §4.2 (baseline justification added).*
>
> ...
>
> (NOTE: we will continue addressing the requested changes in the following comment due to length limitations.)

---

> ### Author Response · Authors · 2026-04-13
> **Response to Reviewer 6GP7 (2/2)**
>
> **2. Multi-seed variance / statistical validation.**
> For latency measurements, Figure 7 now shows ±1σ bands from `blocked_autorange` (GPU timings exclude `torch.compile` warmup).
>
> On the accuracy side, the calibration procedure is deterministic given fixed data and model weights, so there is no seed-dependent randomness to average over. Figure 4(b) already demonstrates robustness to the main source of variation in our pipeline—the calibration dataset: sweeping batch counts from 10 to 80 and epochs from 10 to 80 yields accuracy within 0.25 pp of the optimum. None of the baselines we compare against (ASVD, SVD-LLM, SVD-LLM V2, Basis Sharing, AAFM, GFM) report multi-seed results either, as they share the same deterministic post-training regime.
>
> *Changes: Figure 7 regenerated with ±1σ bands.*
>
> **3. Formal details on β, τ, r.**
> β is now defined with concrete examples in §2.3 (e.g., DeiT-B: [4,4,4]). τ is explained in §3. r is given in closed form by Equation 2 (§2) and is uniquely determined by the budget, group size, and sparsity—it is not tuned.
>
> *Changes: §2.3, §3.*
>
> **4. Training costs.**
> We have made training costs more prominent in the revised manuscript. ViT calibration (block-wise error minimization on 2,560 ImageNet images for 20 epochs) completes in under 1 hour on a single NVIDIA A6000 (§4.1). LLM compression (block-wise optimization for 40 epochs on SlimPajama tokens, no full fine-tuning) completes within 10 hours per model on a single NVIDIA A100 (§4.2). To our knowledge, none of the compared baselines (ASVD, SVD-LLM, SVD-LLM V2, Basis Sharing, AAFM, GFM) report end-to-end compression time in their publications, making FiPS the only method in these comparisons that provides explicit cost figures.
>
> *Changes: §4.1, §4.2.*
>
> **5. Calibration size vs performance, runtime vs competitors.**
> The sensitivity analysis (Figure 5) shows the calibration data trade-off and is now in the main text (§5). We could not find runtime numbers for the baselines to compare against.
>
> *Changes: Figure 5 promoted to §5.*
>
> **6. Attention layers or rescope claims.**
> Claims rescoped to MLPs throughout. We note in §7 that attention projections have the same block-repeated structure, making them a natural next step.
>
> *Changes: abstract, §1, §7.*
>
> **7. Rank-sparsity trade-off, component removal ablation.**
> We believe these ablations are already present in the paper, though perhaps not framed as a single "component-removal" table:
>
> - **Without sparsity (dense V)**: Table 5 reports the "Dense" row at every parameter budget. At 25% budget, accuracy collapses from 80.64% to 65.71% (a 14.9 pp drop). At 10% budget it falls to 15.35%. This is also visible in Figure 3(b), where the dense variant diverges sharply from the sparse ones as group size increases.
> - **Without SVD initialization (random init)**: The RI ablation in §5 shows a 1 pp accuracy drop. Figure 5(a) compares initialization strategies directly.
> - **Without cross-block sharing (no grouping)**: Figure 3(a) and 3(b) show reconstruction error and accuracy as a function of group size, starting from group size 1 (no sharing). Performance improves with sharing up to the optimal group size of 4.
>
> The rank-sparsity trade-off is shown in Figure 2(a): at a fixed 25% budget, increasing sparsity on **V** allows a higher rank, reducing reconstruction error until ~80% sparsity. We also promoted Table 5 (sparsification methods) from the appendix into §5 to make this evidence more accessible.
>
> **8. Reduce reliance on appendices.**
> Moved the sparsification method comparison table and the sensitivity analysis figure from the appendix into §5.
>
> *Changes: §5.*

---

### Review · Reviewer_dqCR · 2026-03-29

**Summary Of Contributions:**

The paper proposes FiPS, a transformer compresion method that combines cross-layer parameter sharing, low-rank decomposition, and sparsity. The main idea is to represent grouped MLP weights across layers using a shared basis and sparse layer-specific projection matrices, initialized with SVD and optimized using block-wise reconstruction losses. Empirically, the method is evaluated on both ViTs and LLMs, where it often preserves accuracy/perplexity better than the selected low-rank baselines at comperable compression levels, and it is also shown to combine with QAT for more aggressive compression.

**Audience:**

Yes

**Audience Explanation:**

Yes. Model compression for transformers remains an active and practicaly relevant area, and this paper studies a less standard direction: fine-grained parameter sharing across layers rather than only per-layer low-rank compression or quantization. The paper is also likely to interest readers working on efficeint inference, sparsity, post-training compression, and deployment of ViTs and LLMs, especially because it evaluates the approach across both modalities and examines how sparsity interacts with shared low-rank factors. Even readers who are not fully convinced by the final method may still find the analysis of concatenation strategy, grouping, and sparsification choices useful.

**Claims And Evidence:**

Yes

**Claims Explanation:**

The paper provides a reasonably broad empirical evaluation across both vision and language models, including DeiT-B, Swin-L, Llama-7B, and Llama-3.1-8B, and reports results over multiple compresion budgets rather than only a single operating point. The evidence for the core methodological claims is strengthened by ablations on sparsification strategy, initialization, grouping, and structured sparsity, as well as by latency maesurements for DeiT-B under structured sparsity. In particular, the main paper and appendix support the claim that sparsifying the projection factor is important, and that the resulting method can outperform dense low-rank baselines under the authors’ chosen setup.

That said, I do not think the empirical case is fully complete. The scope is restricted to MLP compression, so the work does not yet constitute a full transformer compression framework. Some of the strongest claims around practicality would also benefit from more direct evidence on LLM inference speed, not just ViT latency, and the QAT section is narrower than the rest of the paper. In addition, the novelty claim would be more convincing if the relationship to prior cross-layer sharing methods were positioned more clearly and compared more systematically in the main paper. So overall I find the evidence sufficient to support the paper’s central empirical claims, but not all broader claims are equally well substantiated.

**Requested Changes:**

The paper is promising, but I would request the following revisions before publication:

Clarify the paper’s novelty relative to prior cross-layer parameter sharing methods much more explicitly in the main text, not only in rebuttal-style discussion. A direct, prominently placed comparison is requird.

Be more precise about the scope of the contribution: the current method compresses only MLP layers. This is still useful, but the frameing should avoid implying a complete transformer compression solution.

Improve the discussion of practical efficiency. The DeiT latency study is helpful, but the paper should either include LLM-side speed measurements or clearly narrow claims about infrence benefits to the settings that were actually benchmarked.

Strengthen the experimental positioning by clarifying why the selected baselines are the right ones for this setting, and by making sure the comparison to closely related sharing-based approaches is visible in the main experimental section.

Improve presentation clarity: standardize the terminology around “parameter budget” versus “compression ratio,” make the grouping hyperparametere easier to locate, and ensure figures and appendix references are easy to follow. These issues do not invalidate the work, but they currently make the paper harder to read than necessary.

---

> ### Author Response · Authors · 2026-04-13
> **Response to Reviewer dqCR**
>
> We thank the reviewer for their careful reading and constructive suggestions. Below addresses the discussion points and the requested changes.
>
> ### Discussion
>
> On **scope**, all claims now refer to MLP compression specifically. The abstract says "a unified framework for compressing transformer MLPs," and §7 frames attention as future work.
>
> On **LLM efficiency**, we opted to add a FLOP analysis in §4.1 rather than new LLM latency benchmarks (which would require substantial additional compute). The analysis shows the per-token cost reduction (2dp → 2rp(1−s) + 2dr) depends only on the factorization parameters, not on whether the layer sits in a ViT or LLM.
>
> On **novelty**, §6 now explicitly differentiates FiPS from ALBERT, MiniViT, and Basis Sharing, with a summary paragraph laying out what FiPS does differently.
>
> The reviewer also notes that the QAT section is narrower than the rest of the paper. We agree; QAT is meant as a demonstration of orthogonality, not a standalone contribution.
>
> ### Requested Changes
>
> **1. Clarify novelty vs prior sharing methods.**
> §6 now has explicit differentiation and a summary paragraph with FiPS's three distinguishing properties.
>
> *Changes: §6 restructured.*
>
> **2. Scope to MLPs.**
> Abstract and §1 now say "transformer MLPs." Attention is future work (§7).
>
> *Changes: abstract, §1, §7.*
>
> **3. LLM efficiency needs benchmarks or narrower claims.**
> Added FLOP analysis in §4.1. Claims narrowed to the settings we actually benchmarked.
>
> *Changes: §4.1.*
>
> **4. Justify baseline selection.**
> FiPS is a post-training, layer-wise compression method based on matrix factorization. We selected baselines that match this regime so that performance differences reflect the factorization and sharing strategy rather than the benefit of auxiliary training signals.
>
> For ViTs, AAFM (Tukan et al., 2023) performs layer-wise PCA-based compression with MSE error correction, which is directly analogous to FiPS's block-wise error minimization. GFM is its extension with full end-to-end fine-tuning, analogous to our optional FiPS+FT. These are the two closest ViT compression baselines in the literature that share our post-training, factorization-based setting without distillation or LoRA.
>
> For LLMs, ASVD (Yuan et al., 2024), SVD-LLM (Wang et al., 2024b), SVD-LLM V2 (Wang et al., 2024c), and Basis Sharing (Wang et al., 2024a) are all recent, strong layer-wise matrix factorization and sharing baselines published at top-tier AI venues. Together they represent the current state of the art in post-training factorization-based LLM compression.
>
> Methods that rely on knowledge distillation or LoRA introduce auxiliary supervision signals or learned adapters on top of the compression, which makes it difficult to isolate the contribution of the compression strategy itself. We show that FiPS is orthogonal to such techniques by combining it with QAT in Table 4. We are open to including additional baselines if the reviewer can suggest a specific method that operates in the same post-training, layer-wise regime without auxiliary supervision and has publicly available code for the models and compression ratios we evaluate. We would do our best to accommodate this within the revision timeline.
>
> *Changes: §4.1, §4.2.*
>
> **5. Standardize terminology; improve presentation clarity.**
> We keep both "parameter budget" and "compression ratio" because they mean different things: the budget is the controlled input (fraction of MLP params, same across models), while the compression ratio is the output (whole-model reduction, model-dependent via Equation 1). Both are now defined in §2, and Table 1 includes a dedicated row mapping each parameter budget to its approximate whole-model compression ratio. We also expanded β with examples in §2.3, fixed stale appendix references, and added equation labels.
>
> *Changes: §2, §2.3, Table 1 caption, Equation 2.*

---

### Review · Reviewer_MvYU · 2026-04-07

**Summary Of Contributions:**

This work studies compression of transformer models in Large Neural Networks. The authors propose a novel compression approach that integrates three methods in the compression literature: Parameter Sharing, Low-Rank Decompositions, and Sparsity. Namely, the Fully Connected (FC) Layers in Multilayer Perceptrons (MLPs) are concatenated into a short/fat matrix, and the compression method is initialized via a sparse low-rank approximation of the matrix - found via Singular Value Decomposition (SVD) and a separate sparsifying function. The proposed approach is termed Fine-grained  Parameter Sharing (FiPS).

A numerical analysis on several Large Neural Networks shows that FiPS can achieve compression-accuracy results that rival or exceed several state-of-the-art approaches.

**Audience:**

Yes

**Audience Explanation:**

I believe many researchers working on compression, quantization and even low-rank/sparse machine learning/statistical inference would find this work interesting.

**Broader Impact Concerns:**

No Broader Impact Concerns.

**Claims And Evidence:**

Yes

**Claims Explanation:**

The authors claim 4 contributions:

1) Analysis of basis sharing strategies: I believe the authors have fulfilled this. A systematic analysis of various sharing methods was completed, and the reconstruction errors with various levels of sparsity was reported. The experimental results showed that the most effective sharing strategy for effective compression was "long-axis concatenation".

2) FiPS algorithm: It is true that the authors proposed a block/alternating minimization style algorithm to estimate a shared basis and projection matrices. However, I believe the main issue lies in the language used. Such as, declaring the estimated bases as "low-rank" (without further discussion into how optimum ranks are chosen, or how the algorithm that solves for a basis sans low-rank constraint can ensure low-rankness). Or claiming that the reconstructed basis and projection matrix follow a "sparse tensor decomposition", when the work clearly studies the matrix decomposition $W=UV$. In fact, this resembles more a sparse coding/ dictionary learning problem.

3) Comparison with state-of-the-art compression: I believe the authors have fulfilled this. Experiment show that FiPS effectively compressed in vogue datasets (DeiT-B, Swin-L, Llama-7B), with minimum performance loss, and a variety of compression budgets.

4) Quantization-Aware Training (QAT): Same as point 3.

In summary, I believe point 2, FiPS algorithm has not been fully addressed. I suspect the authors are conflating the low-rank *initialization* of FiPS with the *structural property* of the final result (see more below).

**Requested Changes:**

# Major Comments/ Requested Changes.
1) Usage of the term "tensor-decomposition": I think describing the paper's proposed approach as a tensor-based approach, particularly in the title, is misleading to the readers and factually inaccurate. FiPS consider a matrice $W \in R^{d\times p}$, and outputs basis $U$ and projection $V$. I do not see any tensor-specific operations or algebra in the approach. Additionally, the paper does not indicate any kind of simple or intuitive extension of their work to the true tensor case (a multidimensional array with 3 or more dimensionalities/modes).

2) Achieving "low-rank" transformer decompositions: I am a little concerned whether the basis $U$ in FiPS is indeed a low-rank orthogonal basis. I believe having the answers to the following questions/ comments may help:
  a) Authors initially claim that $r<p$ yet $p>d$. Is this not an empty claim as $r=d$ is possible and therefore the problem is no longer meaningfully low-rank?
  b) Are validation experiments performed to find the best rank $r$ in FiPS? If so, how are these experiments conducted (Ad Hoc, grid search etc)?  Additionally Does the chosen rank remain stable or constrained during the optimization process?
  c) Are the authors claiming that the basis $U$ is an orthogonal basis? When the resulting rank, $r$, is found to be greater than the model size, the proposed approach is to enlarge the basis $U$. $U$ is now an over-complete basis and is neither low-rank nor orthogonal. I do not believe the solution to this issue, shared in Section 3, is clear or convincing. Can the authors show, through mathematical derivation or a specific example, FiPS ensures a low-rank and/or orthogonal basis, $U$ (if that is the claim)?
  d) Equation $1$ shows the minimization function for the Local Error Minimization Step in FiPS. I understand that Equation $1$ is solved via a Block Coordinate Descent approach, however, there is no inclusion of constraints for projection matrices $V_i$ and basis $U$ (such as a nuclear norm regularization for low-rankness, $\ell_1$ norm for sparsity low-rankness and separate orthogonality constraint). The sparsifying step for $V_i$ is applied before the Local Error Minimization Step, which will estimate a $V_i$ that may or may not be sparse. How are the authors ensuring that such constraints are fulfilled?

# Minor Comments:
1) MLP is not defined throughout the paper.
2) I think it would be useful for the reader if the error and accuracy metrics of the experiments were formally defined through mathematical formulas.

---

> ### Comment · Action_Editor_Eq96 · 2026-04-07
> **Please update your review**
>
> Dear Reviewer MvYU,
>
> Thank you for submitting the review.
>
> I feel that there is a discrepancy between your `Major Comments` and `Are the claims made in the submission supported by accurate, convincing and clear evidence?: Yes`. Is your answer to the question actually `No`?
>
> Could you also please clarify
> 1. what claims made in the submission are (or NOT) supported by accurate, convincing and clear evidence and
> 2. what the evidence is
> ?
>
> Thank you,
> Your AE

---

> > ### Comment · Reviewer_MvYU · 2026-04-09
> > **Review Update**
> >
> > Dear AE,
> >
> > Please refer to the edited review above.
> >
> > thank you,

---

> > > ### Comment · Action_Editor_Eq96 · 2026-04-09
> > >
> > > Dear Reviewer MvYU,
> > >
> > > Thank you for updating the review!
> > >
> > > Best,
> > > Your AE

---

> ### Author Response · Authors · 2026-04-13
> **Response to Reviewer MvYU**
>
> We thank the reviewer for their time and comprehensive feedback. We summarize below the discussion points and address the requested changes.
>
> ### Discussion
>
> **(a) r > d is possible, so the problem is not meaningfully low-rank.**
> Yes, that is correct. We agree with the reviewer's observation that FiPS in this regime resembles sparse dictionary learning. In the revision we now distinguish two regimes in §2: for r ≤ d this is a standard low-rank factorization, but for r > d, **U** becomes an overcomplete shared dictionary and the compression comes from sharing and sparsity, not from rank reduction. We adopt this terminology explicitly in §3 and no longer claim low-rankness when r > d.
>
> **(b) How is the rank r selected? Does it remain stable during optimization?**
> r is not something we tune. For a group of N layers with weight matrices in ℝ^{d×p}, the total nonzero MLP parameters is d·r (for **U**) plus N·(1−s)·r·p (for the sparse **V**_i's). Setting this equal to the parameter budget gives:
>
> r = (budget · N·d·p) / (d + N·(1−s)·p)
>
> So r is uniquely determined by the budget, group size N, sparsity s, and layer dimensions (d, p). Once computed at initialization, r stays fixed throughout optimization. This closed-form expression is now given as Equation 2 in §2, immediately after the compressed-size formula.
>
> **(c) Is U claimed to be orthogonal? It becomes overcomplete when r > d.**
> When r ≤ d, **U** is orthogonal at initialization because it comes from SVD; we now state this explicitly at first mention in §2. However, we do not constrain **U** to stay orthogonal during optimization. When r > d, **U** is overcomplete and cannot be orthogonal; additional dimensions are initialized with dampened singular values (via τ > 1) so they do not dominate early on. Both regimes are now clearly distinguished in §2 and §3.
>
> **(d) Equation 2 has no explicit low-rank, sparsity, or orthogonality constraints.**
> The sparsity structure in **V** is maintained by the GMP schedule, which prunes and regrows entries each epoch. **U** is learned freely with no constraints. We intentionally omit nuclear norm or orthogonality regularization because we do not need **U** to remain low-rank or orthogonal after initialization. The empirical justification is in Table 5 (§5): the "Dense" row shows that removing sparsity collapses accuracy by 14.9 pp at 25% budget, confirming that GMP alone is sufficient to enforce the structure we need.
>
> ### Requested Changes
>
> **1. Usage of "tensor decomposition."**
> We respectfully disagree that the term is inaccurate. A matrix is a 2nd-order tensor, and SVD is the order-2 special case of Tucker decomposition (Kolda & Bader, 2009). More importantly, the concatenated weight collection $\mathcal{W} = \{\mathbf{W}_1, \ldots, \mathbf{W}_N\}$ is naturally a 3rd-order tensor $\mathcal{W} \in \mathbb{R}^{d \times p \times N}$; the long-axis concatenation used in FiPS is its mode-2 unfolding, and the resulting factorization into a shared $\mathbf{U}$ and layer-specific sparse $\mathbf{V}_i$ corresponds to a Tucker-1 decomposition with sparse core slices. SVD is widely classified as a tensor decomposition in the literature, regardless of the order of the operand.
>
> We acknowledge, however, that the term could set an expectation of higher-order tensor algebra (tensor train, tensor ring, CP) that is not what FiPS employs. To make this distinction clear, we have added a sentence in §2 noting that FiPS operates via the mode-2 unfolding of the weight tensor and that the decomposition is the order-2 (matrix) case of Tucker-1. We have retained **"Sparse Tensor Decomposition"** in the title as it is mathematically precise, while ensuring the text does not overstate the connection to higher-order methods.
>
> **2. "Unified framework" but only MLP layers evaluated.**
> We clarified that "unified" refers to combining sharing, factorization, and sparsity in one optimization, not to covering all transformer components. The abstract now says "transformer MLPs." Extending to attention is discussed as future work in §7.
>
> **3. MLP not defined.**
> Fair point. MLP is now defined at first use in the abstract: "Multi-Layer Perceptrons (MLPs)," and again in §1 for readers who skip the abstract.
>
> **4. Accuracy and error metrics should be formally defined.**
> We added an Evaluation Metrics paragraph at the start of §4 that defines top-1 accuracy, perplexity (with the formula), and MSE. All equations are now properly labeled and cross-referenced.

---

### Comment · Action_Editor_Eq96 · 2026-04-07
**Start author-reviewer discussions**

Dear authors,

You have three reviews. Please address their comments as soon as possible. The discussion period will end on April 20th, 2026.

---

Dear reviewers,

Thank you for submitting your reviews. As this paper received three reviews, now it's time to have discussions with authors.

You will be able to submit your formal decision recommendation starting in 2 weeks. Your prompt responses to authors' comments would be very appreciated.

Remember that different from other journals / conferences, [TMLR's acceptance criteria](https://jmlr.org/tmlr/reviewer-guide.html) are based on positive answers to the following two questions

- Are the claims made in the submission supported by accurate, convincing and clear evidence?
- Would some individuals in TMLR's audience be interested in the findings of this paper?

Novelty of the studied method is not a necessary criteria for acceptance at TMLR.

It's still not too late to review [the TMLR's reviewer guideline](https://jmlr.org/tmlr/reviewer-guide.html) in case you are not aware of that.


Thank you!
Your AE

---

### Author Response · Authors · 2026-04-13
**Meta Response to Reviewers**

We thank the Action Editor and all three reviewers for their prompt and constructive feedback. All three reviewers found the claims supported by evidence and the topic of interest to the TMLR audience. The reviews converge on several themes: (i) the title and terminology should clarify the relationship between the factorization used in FiPS and higher-order tensor methods, (ii) claims should be scoped to MLP compression rather than implying a full transformer compression solution, (iii) baseline selection and practical efficiency need clearer justification, and (iv) key implementation details should be more accessible in the main text. We have addressed all of these in the revised manuscript. The major changes include, but are not limited to:

- **Title and terminology:** Retained "Sparse Tensor Decomposition" in the title, as the term is mathematically accurate: FiPS operates on the mode-2 unfolding of a 3rd-order weight tensor and its factorization corresponds to Tucker-1 with sparse core slices. Added clarifying language in §2 to distinguish this from higher-order methods (tensor train, tensor ring, CP).
- **Scoped claims:** The abstract, introduction, and conclusion now explicitly refer to transformer MLP compression. MLP is now defined at first use in the abstract. Extending to attention layers is discussed as future work.
- **Algorithmic clarity:** Added a closed-form expression for the rank $r$ (Equation 2), clarified the low-rank vs. overcomplete regimes, and provided formal definitions of all evaluation metrics.
- **Stronger main text:** Promoted the sparsification method comparison and the sensitivity analysis from the
appendix into §5, and added explicit training cost figures and a FLOP analysis.
- **Positioning:** Expanded the related work (§6) with explicit differentiation from ALBERT, MiniViT, and Basis Sharing, and added baseline justification paragraphs in §4.
- **Statistical reporting:** Added ±1σ variance bands to all latency benchmarks (Figure 7).
- **Compression ratios:** Added compression ratio rows to Tables 1, 2, and 5, mapping each parameter budget to its approximate whole-model compression ratio. Abstract claims updated to use these verified ratios.
- **Equation placement:** Moved the rank equation (Equation 2) from §3 to §2 for earlier accessibility.

We respond to each reviewer individually in separate comments.

---

### Comment · Action_Editor_Eq96 · 2026-04-22
**Submit your official recommendation**

Dear Reviewers,

Now it's time for you to submit an official recommendation for this submission.

The authors responded to your initial reviews.

Please
1. consider their rebuttal comments,
2. respond to the comments ***ASAP*** if you have follow-up questions or comments, and
3. submit your official recommendation ***by May 5th***, including ***your thoughts on the author rebuttal***, specifically, what point in your review their responses addressed AND did not address. It will greatly help me write a meta-review and make a fair decision.

Thank you,
Your AE

---

### Decision · Action_Editor_Eq96 · 2026-05-14

**Recommendation:** Accept as is

**Audience:**

Yes

**Audience Explanation:**

Empirical discussions on model compression, quantization, and low-rank/sparse machine learning should be an interest of TMLR's audience.

**Claims And Evidence:**

Yes

**Claims Explanation:**

All reviewers indicated that the claims made in this work are supported by accurate, convincing and clear evidence.
Some of them were initially not satisfied with evidence and/or misalignment between evidence and claim, but they confirmed that those were addressed through the rebuttal.